# Association-induced folding governs surrogate light chain and pre-B cell receptor core assembly

Jasmin König[1,2], Natalia Catalina Sarmiento Alam[1,2,4], Ruiming He[1,2],
Nicolas Blömeke[1,2], Olga Sieluzycka[1], Florian Rührnößl[1,2], Maximilian Riedl [1,2],
Bernd Reif [1,3], Matthias J. Feige [1,2] & Johannes Buchner [1,2] ✉

Binding of the surrogate light chain (SLC) to the heavy chain (HC) of the pre-B cell receptor (preBCR) is an important quality control checkpoint during B cell development as roughly 50% of the rearranged HCs are defective. Unlike the regular light chain (LC), the SLC is a hetero-dimer of VpreB and λ5, both containing unstructured extensions, the unique regions. The molecular mechanisms that underlie the complex assembly processes which give rise to the final pre-BCR is not fully understood. Here we show, via reconstitution of the pre-BCR in vitro and in cells that λ5 plays a key role in the pre-BCR assembly. During SLC assembly, a β-strand, located between the λ5 domain and the unique region, induces structure in the largely unfolded VpreB, creating a high affinity complex. In addition, association of λ5 with the unstructured HC $C_H1$ domain is required for its folding. This is essential for pre-BCR assembly and its release from the endoplasmic reticulum (ER). Finally, the unique region of λ5 plays a pivotal role in the antigen interaction of the SLC-HC complex. Together, our results reveal a multi-step mechanism for SLC and pre-BCR assembly, governed by association-induced folding reactions required for structural integrity and function.

Antibodies are a cornerstone of the humoral adaptive immune system. They fulfill important functions in protecting the body against foreign molecules and pathogens. The production of antibodies critically depends on the successful progression of hematopoietic stem cells to mature B cells[1]. During this process, developing B lymphocytes have to pass multiple stages and checkpoints, in which the genes for heavy chains (HC) and light chains (LC) of immunoglobulins undergo recombination and scrutiny[2] to produce a unique immunoglobulin sequence in each mature B cell[3].

HC and LC gene rearrangements take place sequentially[4]. First, HCs are rearranged and expressed in the absence of LCs, in earlier developmental stages, in a membrane-bound form[5]. Thus, a key

checkpoint during B-cell development is to monitor whether the rearrangement of the HC gene locus resulted in a functional HC protein. This is validated by the ability to productively associate with the so-called surrogate LC (SLC)[6–9], often also called invariant LC. The assembly of the SLC and the HC together with the CD79a/CD79b signaling components forms the pre-B cell receptor[10–12] and induces an intracellular signaling cascade that promotes LC gene rearrangement and downregulation of the expression of λ5 and VpreB, the components of the SLC[13–17]. This is a critical step where approximately 50 % of cells that produce a certain HC are discarded because they do not form a productive complex with the SLC[18,19]. Thus, the SLC has an essential function in B-cell development and antibody production.

[1]Department Bioscience, School of Natural Sciences, Technical University Munich, Garching, Germany. [2]Center for Functional Protein Assemblies, Technical University Munich, Ernst-Otto-Fischer-Strasse 8, Garching, Germany. [3]Institute of Structural Biology (STB), Helmholtz-Zentrum München (HMGU), Ingolstädter Landstr. 1, Neuherberg, Germany. [4]Present address: Sanofi, Technologiepark 21 9052 Ghent/zwijnaarde, Ghent, Belgium. ✉e-mail: johannes.buchner@tum.de

The SLC is structurally similar to a regular LC but also shows key differences. It consists of two proteins, which are non-covalently associated: VpreB, which corresponds to the variable LC domain, and λ5, which corresponds to the constant domain of a conventional LC of the λ-subtype[20–22]. VpreB interacts with the $V_H$ domain of the HC, and λ5 interacts with the HC $C_H1$ domain[20,21,23]. The association of the SLC with the HC involves both non-covalent interactions and a covalent intermolecular disulfide bond between λ5 and the $C_H1$ domain[3,24–28]. VpreB has the typical V-type Ig-fold but contains only 8 out of 9 β-strands of a conventional $V_L$ domain. It lacks a J region that forms the ninth β-strand[3,20,21]. The missing strand is provided by λ5, which comprises in its folded domain the seven β-strands of a conventional $C_L$ domain and, in a sequence extension outside the folded domain, the additional β-strand missing in VpreB[3]. This additional β-strand is important for the association of λ5 and VpreB[29] by a process called β-strand complementation[30].

Further major differences to regular LCs are the so-called unique regions (UR) of VpreB and λ5. These are unstructured extensions with sequences different from any known protein. The UR of VpreB is 24 aa long and is located at the C-terminal end of the domain, whereas the folded domain of λ5 contains an additional β-strand and a 50 aa long UR at its N-terminus[21]. The URs are not required for SLC assembly[29]. However, the URs of VpreB and *λ5* may replace the missing complementarity-determining regions (CDR) of VpreB[21].

In the available crystal structure of the pre-BCR, the URs are only partially resolved[21]. It thus remains unclear what their specific contribution to SLC assembly and function is and whether they affect antigen interaction. Furthermore, despite progress in understanding structural aspects and the biological function of the SLC, little is known about the assembly process of VpreB and λ5 and, especially, the formation of the complex of the SLC with the HC, i.e., the pre-BCR. Understanding these processes is of high biological relevance because of the checkpoint function of the SLC. To address these open questions, in this study, we have reconstituted and analyzed the assembly processes leading to the SLC and the SLC-HC complexes in vitro and in mammalian cells. Our results resolve the underlying mechanisms and provide insight into the molecular quality control function of the SLC on HCs as a key step in generating a functional pre-BCR.

## Results

### Formation of the SLC heterodimer involves assembly-mediated folding of VpreB

While the structure of the SLC in complex with a HC fragment has been determined (Fig. 1a)[21], we know little about the structure of the two individual SLC proteins, VpreB and *λ5*, and the association reactions leading to the SLC and the SLC-HC complex. To address these questions, we analyzed these processes in cells and in vitro using the wildtype proteins and constructs devoid of characteristic SLC regions, the extensions called unique regions (URs) or the extra β-strand in *λ5* (Fig. 1b).

When expressed individually in HEK293T cells, both subunits of the SLC were weakly secreted into the cell medium (Fig. 1c). Co-expression of the proteins increased their secretion (Fig. 1c), in agreement with previous studies[29], concomitant with O-glycosylation of VpreB (Supplementary Fig. 1a). Furthermore, the overall levels of λ5 and VpreB increased upon co-expression, arguing for mutual stabilization in cells upon assembly (Fig. 1c). In conclusion, λ5 and VpreB on their own seem to be partially recognized as non-native by the endoplasmic reticulum (ER) quality control machinery and are thus partially retained in the ER. When they associate as the SLC, they seem to be no longer a target for the ER quality control and are thus efficiently transported out of the cell.

To gain further insights into the underlying mechanisms, we studied the structure and stability of the individual SLC proteins in vitro. The far-UV circular dichroism (CD) spectrum of λ5 (Fig. 1d) revealed

that the isolated protein is characterized by β-strands and unfolded segments consistent with its structure in the SLC complex[21] and the presence of an unstructured extension. In contrast to the structure of VpreB in the SLC complex, the far-UV CD spectrum of isolated VpreB (Fig. 1d) indicated that the protein is largely unfolded under the conditions tested. In agreement with this notion, nuclear magnetic resonance (NMR) spectroscopy revealed that the $^{15}N$-$^1H$ HSQC spectrum of isolated VpreB is characteristic of an unfolded protein (Fig. 1e, red spectrum). Although VpreB shares homology with the $V_L$ domain of a conventional light chain, it lacks a conserved β-strand and therefore cannot fold independently into a stable Ig-like domain. Importantly, conventional $V_L$ domains cannot replace VpreB in the pre-BCR, showing that VpreB and λ5 together provide unique structural features required for pre-BCR assembly. When unlabeled λ5 was added to isotope-labeled VpreB (Supplementary Fig. 1b, orange spectrum), the NMR spectrum of VpreB became well-resolved in agreement with association-induced folding of VpreB by λ5. Interestingly, also a peptide (λ5β) representing the additional ß-strand donated by λ5 was able to induce folding of VpreB (Fig. 1e, blue spectrum). Many of the dispersed peaks in the spectrum of this complex superimpose exactly with the resonances observed upon addition of λ5 (compare Supplementary Fig. 1b).

To analyze the conformational changes in VpreB and λ5 upon SLC formation further, hydrogen-deuterium exchange coupled to mass spectrometry (H/DX-MS) was performed (Fig. 2a, b). For VpreB, the H/D exchange and thus the conformational flexibility decreased in the SLC complex, as upon binding of the β-strand, VpreB adopts its native structure (Fig. 2a). In agreement with this, the additional β-strand in λ5 became highly protected against exchange in the SLC complex (Fig. 2b). To a lesser extent, also in λ5, a generally increased protection against H/D exchange was observed upon assembly with VpreB, which implies that upon complex formation both VpreB and λ5 undergo structural rearrangements and/or become less dynamic (Fig. 2b). Furthermore, both URs show increased protection against H/D exchange in the SLC complex compared to the individual proteins. This is unexpected since the URs were reported to be largely unstructured and protruding from the complex[21]. The kinetics of VpreB folding upon association with λ5 monitored via changes of the CD signal (Fig. 2c and Table 1) revealed a time constant of τ = 51.0 min at 25 °C, in agreement with the pronounced conformational rearrangements in VpreB and λ5 upon formation of the SLC as described above. Taken together, VpreB is an unfolded protein that can only fold in the presence of the β-strand provided by λ5. The association is accompanied by conformational changes in both proteins involving the URs.

The benefit of the intricate complementation of the VpreB fold upon association with λ5 is that the folded state of VpreB is only attained in the heterodimeric SLC complex. Furthermore, the strand insertion reaction potentially allows creating a high-affinity complex. In agreement with this idea, we determined the dissociation constant ($K_D$) of the SLC complex to be 17.7 nM (Fig. 2d and Table 1). Association with VpreB slightly reduces the stability of λ5 (Fig. 2d and Table 1). Interestingly, for the SLC in complex with a HC fragment consisting of $V_H$ and $C_H1$ (also referred to as Fd), the $T_m$ is even lower (Fig. 2e and Table 1).

Quaternary structure analysis by AUC confirmed that the SLC complex, reconstituted in vitro, sediments as a heterodimer consisting of VpreB and λ5 (~3.2 S); the λ5 monomer sediments at ~1.8 S (Fig. 2f). For isolated VpreB, the sedimentation coefficient was also ~3.2 S; however, the broad profile compared to folded controls suggests that this signal arises from a heterogeneous mixture of oligomeric species in the range of dimers to tetramers. This interpretation is consistent with previous reports describing non-native dimeric or higher-order VpreB assemblies formed under certain in vitro conditions[30–32]. Thus, the interaction of monomeric λ5 with VpreB results in the dissociation of the VpreB dimer and the formation of the SLC heterodimer. Further

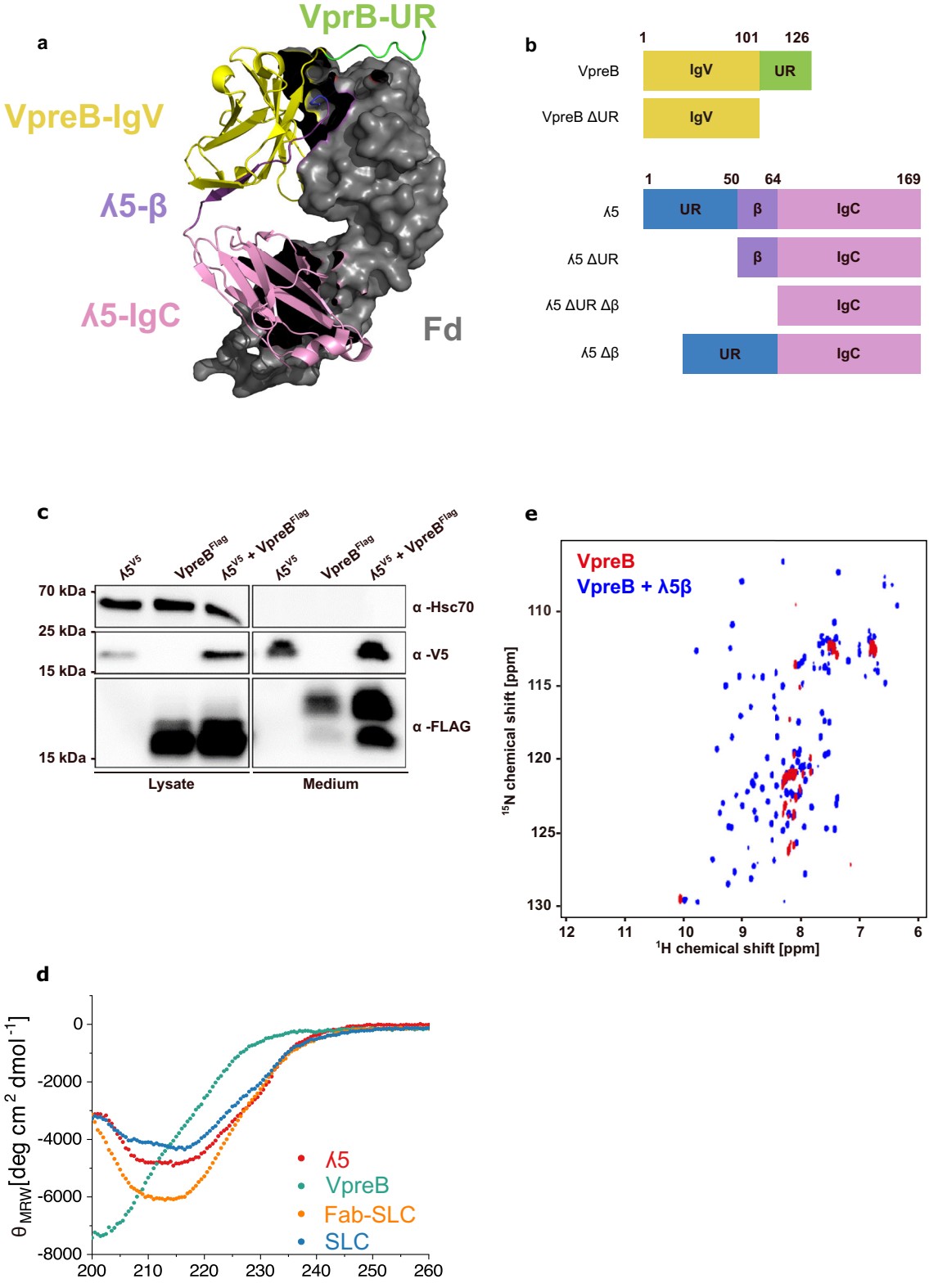

**Fig. 1 | Structure of pre-B lymphocyte protein (VpreB), immunoglobulin lambda 5 (λ5), and the surrogate light chain (SLC). a** Cartoon structure of the SLC in complex with the fragment determinant (Fd) part of the heavy chain (PDB: 2H32), the VpreB domain in yellow, a part of its unique region (UR) is green and the λ5 domain is pink with its additional β-strand in violet. The URs are not completely resolved in the crystal structure. **b** Overview of SLC variants. **c** Expression and secretion of V5-tagged λ5, FLAG-tagged VpreB and both proteins together in HEK293T cells. Hsc70 was included as a positive control for lysed cells. In the

λ5 + VpreB lane of the medium fraction, two bands are observed for VpreB. The lower molecular weight band likely represents non-glycosylated VpreB (see Supplementary Fig. 1a). This experiment was performed twice. **d** Far-UV circular dichroism (CD) spectra of fragment antigen-binding (Fab)-SLC, SLC, λ5 and VpreB. ($n = 3$). **e** $^1$H $^{15}$N correlation spectrum of 100 μM VpreB (red), superimposed with a spectrum obtained from a 100 μM VpreB sample that was incubated with a 200 μM solution of unlabeled λ5β peptide (blue). The spectra were recorded at 600 MHz and the sample temperature was 25 °C.

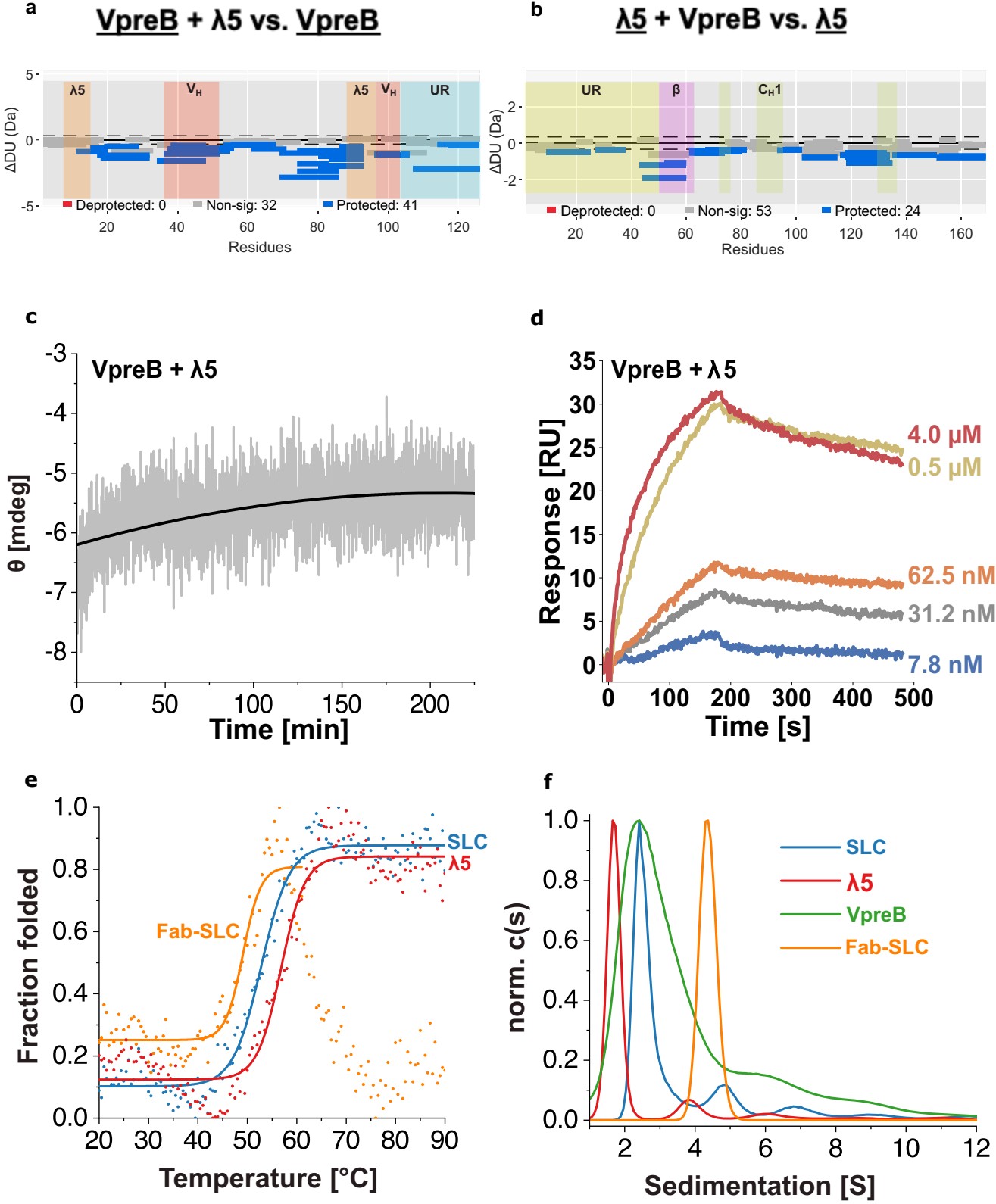

association into the VpreB-λ5-Fd Fab-like complex gives rise to a heterotrimer with a sedimentation coefficient of ~4.5 S (Fig. 2f).

**The unique regions impact folding, stability and association of the SLC**
In addition to the heterodimeric nature of the SLC, a further major difference to regular LCs is the presence of sequence extensions, the

URs, on both proteins. As their functions are largely unknown, we created constructs in which the URs were removed (Fig. 1b) and first analyzed their influence on the structure, association and stability of the SLC proteins. In agreement with the notion that the URs are unfolded, CD spectra of SLC ΔUR and λ5 ΔUR both showed an increased β-strand content, compared to the wildtype, respectively, whereas only minor changes were observed for VpreB ΔUR (Fig. 3a,

**Fig. 2 | Association of immunoglobulin lambda 5 (λ5) and pre-B lymphocyte protein (VpreB). a, b** Woods´s plots displaying the differences in deuterium uptake in VpreB (**a**) and λ5 (**b**) in the surrogate light chain (SLC) complex compared to the single domains. Peptides colored in blue or red, are protected or deprotected from exchange upon forming the SLC, respectively. Peptides with no significant difference (non-sig) upon interaction, determined using a 99% confidence interval (dotted line), are shown in gray. HDX-MS experiments were performed in two independent experiments, each including three technical replicates per time point. HDX-MS data were analyzed using Deuteros, and no formal statistical hypothesis testing was performed. The VpreB unique regions (UR) is colored in blue, the λ5-UR in yellow, the λ5-Interface in orange, the heavy-chain variable domain ($V_H$)-Interface

in red, the additional β-strand in magenta and the constant heavy chain domain 1 ($C_H1$)-Interface in green. **c** Structural changes upon SLC formation were monitored by the change in circular dichroism (CD) signal at 205 nm. A single exponential trace was observed and fitted to an exponential decay function to obtain the folding rate of the reaction. Representative data from $n = 2$ independent experiments are shown. **d** The dissociation constant ($K_D$) and kinetic constants ($k_{on}$, $k_{off}$) for the interaction were determined by SPR and measured in duplicates. The constants are summarized in Table 1. **e** Thermal unfolding transitions of SLC, λ5 and fragment antigen-binding (Fab)-SLC were followed by CD spectroscopy at 205 nm ($n = 2$). **f** Analytical ultracentrifugation (AUC) analysis of λ5, VpreB, SLC and Fab-SLC. Data from a single experiment ($n = 1$) are shown.

**Table 1 | Folding and association kinetics, affinity and thermal stability of immunoglobulin lambda 5 (λ5) and pre-B lymphocyte protein (VpreB) variants and complexes**

| | | $T_m$ [°C] | $K_D$ [nM] | τ [min] | $k_{on}$ [1/Ms] | $k_{off}$ [1/s] |
|---|---|---|---|---|---|---|
| λ5 | | 55.8 ± 0.7 | - | - | - | - |
| SLC | | 53.3 ± 0.4 | - | - | - | - |
| Fab-SLC | | 49.2 ± 0.2 | - | - | - | - |
| λ5 ΔUR | | 58.5 ± 0.3 | - | - | - | - |
| SLC ΔUR | | 56.1 ± 0.5 | - | - | - | - |
| λ5 | VpreB | 53.7 ± 0.7 | 17.7 ± 1.2 | 51.0 ± 3.4 | 35795 ± 2445 | 0.0006 ± 0.0000007 |
| λ5 | VpreB ΔUR | 55.1 ± 0.1 | 3.0 ± 1.3 | 40.9 ± 10.7 | 228450 ± 66850 | 0.0006 ± 0.0001 |
| λ5 ΔUR | VpreB | 56.5 ± 0.8 | 13.5 ± 0.4 | 113.5 ± 29.6 | 89615 ± 2025 | 0.001 ± 0.000009 |
| λ5 ΔUR | VpreB ΔUR | 56.1 ± 0.5 | 5.0 ± 0.8 | - | 321000 ± 111600 | 0.002 ± 0.0003 |

The time constant for folding τ in [min] of VpreB and λ5 variants. The thermal stability (Tm) of the respective complexes or single proteins was assessed by far-UV circular dichroism (CD) spectroscopy at 205 nm. The affinity of the complexes was measured by SPR to determine the $K_D$ in [nM], $k_a$ in [Ms⁻¹], and the $k_d$ in [s⁻¹]. The affinity between VpreB and the β-strand peptide was measured by isothermal titration calorimetry (ITC). Given errors indicate the standard deviations.

compare to Fig. 1d). The analysis of the thermal stability of λ5 ΔUR revealed a $T_m$ of 58.5 °C, which is slightly higher than that of λ5 (Fig. 3b and Table 1).

The λ5-UR also affected the association-coupled folding of VpreB. The time constant τ for folding was determined to be 113.5 min at 25 °C (Fig. 3c and Table 1), which is about two times slower than if λ5 was used (51.0 min). This indicates that the λ5-UR, although dispensable for SLC assembly, accelerates the association-induced folding of VpreB. Furthermore, it slightly destabilizes the SLC complex as reflected in a 3 °C lower Tm (Fig. 3d and Table 1) and a marginally decreased $K_D$ value (Supplementary Fig. 2a and Table 1).

Compared to the results obtained for λ5-ΔUR, the deletion of the VpreB-UR had an opposite effect on the association-induced folding of VpreB. Here the deletion resulted in faster folding compared to the wild-type (WT) complex with a rate constant τ of 40.9 min (Fig. 3e and Table 1) and a stronger binding with a $K_D$ for λ5 and VpreB ΔUR of 3.0 nM, which is sixfold tighter compared to the WT proteins (Supplementary Fig. 2b and Table 1). Also, the thermal stability of this complex is slightly increased by about 2 °C compared to the WT (Fig. 3d and Table 1). Together, these results indicate that the VpreB-UR negatively impacts folding and destabilizes the SLC complex.

Interestingly, in the absence of both URs, a $K_D$ of 5.0 nM was determined for the SLC (Supplementary Fig. 2c and Table 1), which is similar to that of the complex between λ5 and VpreB ΔU (3.0 nM), supporting the idea that mainly the VpreB-UR causes the decreased affinity. The $T_m$ of the SLC ΔUR is about 3 °C higher than that of SLC (56.1 vs. 53.3 °C) (Fig. 3b and Table 1). The AUC analysis revealed SLC ΔUR to be a heterodimer, while isolated VpreB ΔUR remains a homodimer and λ5 ΔUR a monomer (Fig. 3f).

Building on our results on the modulatory effects of the URs on SLC assembly, we asked how deleting the URs affects SLC-Fd secretion from cells. We found that whereas the URs within both, λ5 and VpreB, were not essential for inducing SLC-Fd secretion, deleting the UR of λ5 or VpreB significantly reduced their levels in cells and made the

secretion of the complex less efficient (Fig. 3g). The effect was strongest, if both SLC subunits lost their URs (Fig. 3g). This was also true for a second Fd fragment tested (Supplementary Fig. 2d). These results indicate that the URs play a role in the assembly and stability of the SLC-Fd complex in cells.

In combination, our results reveal complex and differential effects of the two URs on the association and stability of the SLC. Especially the UR of λ5 accelerates the association of the SLC but has a negative effect on the affinity of the complex—and the URs generally seem to be important for secretion from cells.

### The λ5 core region drives folding of $C_H1$

During IgG assembly, the unfolded $C_H1$ domain of the HC folds upon interaction with the $C_L$ domain of the LC[33,34]. It was thus reasonable to assume that the SLC, more precisely λ5, which corresponds to the $C_L$ domain, could also induce the folding of the $C_H1$ domain. To test this hypothesis, we first analyzed the fate of an HC Fd fragment expressed in HEK293T cells in the presence or absence of the SLC proteins. When the Fd fragment was expressed alone, it was not secreted (Fig. 4a), consistent with the unfolded state of the $C_H1$ domain which induces ER retention[29,34]. Co-expression of λ5 and VpreB resulted in the secretion of Fd (Fig. 4a), suggesting that the $C_H1$ domain was folded by the SLC. Expression of the two constituent SLC proteins individually with Fd revealed that specifically λ5 rendered the Fd segment secretion-competent, while VpreB did not allow for Fd secretion (Fig. 4a). The same effects were observed for a different Fd fragment expressed together with the SLC proteins (Supplementary Fig. 3a). To extend these findings, we performed co-immunoprecipitation experiments of Fd with both the SLC and the individual SLC proteins. Our results show that both, λ 5 and VpreB, when expressed in HEK293T cells, can interact individually with Fd, Thus, for VpreB, the lacking β-strand is not a prerequisite for Fd binding. (Fig. 4b and Supplementary Fig. 3b), but only λ5 or the SLC are able to induce Fd secretion (Fig. 4a and Supplementary Fig. 3a).

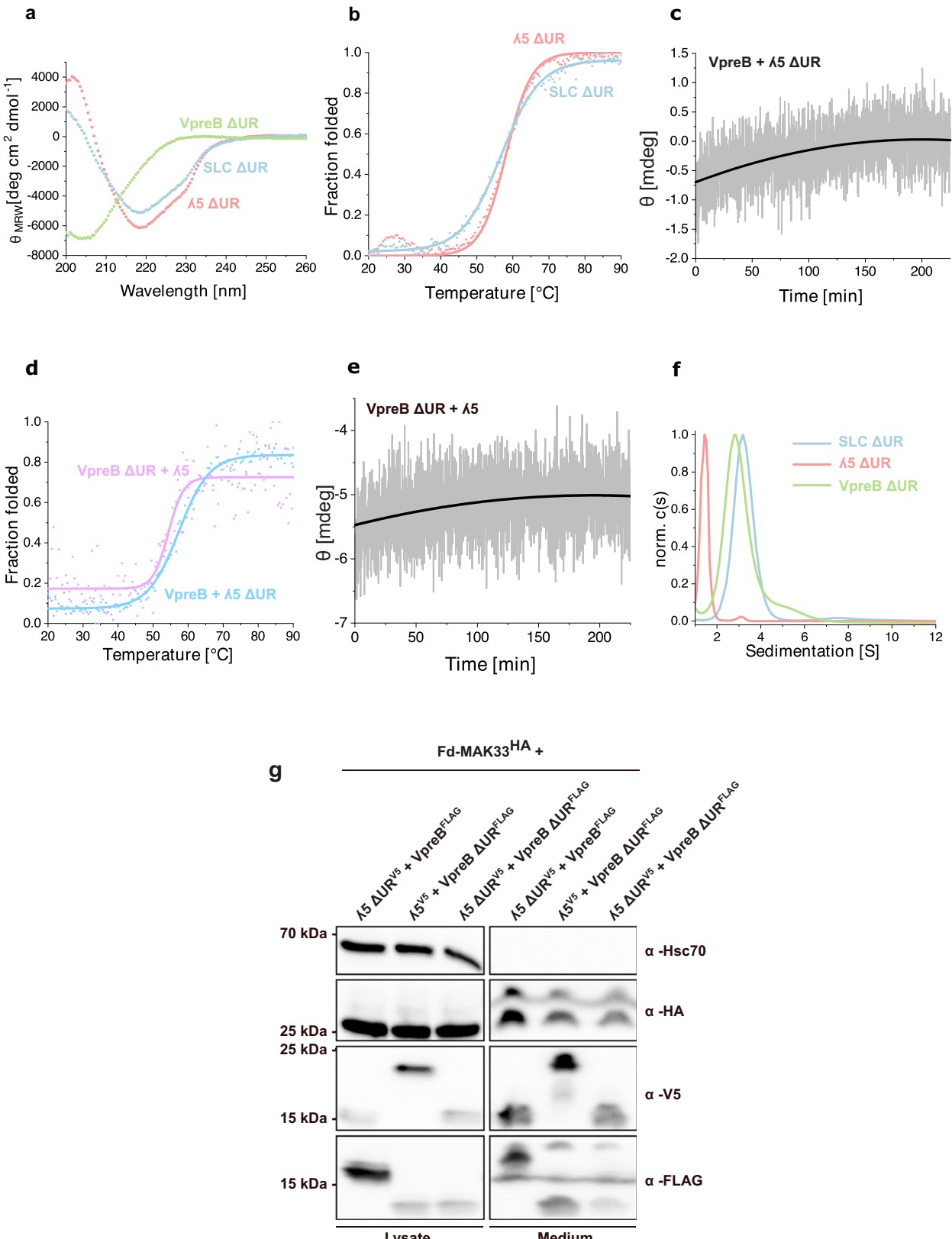

**g** Fd-MAK33$^{HA}$ +

Building on these results, we set out to analyze the potential association-induced folding of the C$_H$1 domain by λ5 in vitro. To this end, we monitored the kinetics of C$_H$1 folding by CD spectroscopy in the presence of the SLC or λ5. When we added the SLC to the unfolded C$_H$1 domain, we observed folding kinetics characterized by a time constant of τ = 53.9 min (Fig. 4c and Table 2). In the presence of λ5, a

time constant of τ = 41.4 min was determined (Fig. 4d and Table 2). As a control, VpreB was added, and here, as expected from the cell-based experiments, no folding kinetics were detected (Fig. 4e). Interestingly, the kinetics for the induced folding of C$_H$1 by λ5 is in the same range as that observed for C$_H$1 and C$_L$ before[34]. The MAK33 C$_H$1 used in our experiments represents an IgG C$_H$1 domain. The pre-BCR is assembled

**Fig. 3 | Role of the unique regions (UR) in the surrogate light chain (SLC). a** Far-UV circular dichroism (CD) spectra of SLC ΔUR, immunoglobulin lambda 5 (λ5) ΔUR, pre-B lymphocyte protein (VpreB) ΔUR. ($n = 3$ independent experiments). **b**−**d** Thermal induced unfolding transitions of λ5 ΔUR or SLC ΔUR (**b**; $n = 3$ independent experiments) as well as VpreB + λ5 ΔUR or VpreB ΔUR + λ5 (**d**; $n = 2$ independent experiments) were followed by CD spectroscopy at 205 nm. **c**− **e** Structural changes upon association were monitored by the change in signal at 205 nm by far-UV CD spectroscopy for VpreB and λ5 ΔUR (**c**) as well as VpreB ΔUR + λ5 (**e**). A single exponential trace was observed and fitted to an exponential decay function to obtain the folding rate of the reaction. Representative traces from $n = 2$ independent experiments are shown. **f** Analytical ultracentrifugation (AUC) analysis of SLC ΔUR, λ5 ΔUR and VpreB ΔUR. Data from a single experiment ($n = 1$) are shown. **g** Expression and secretion of murine fragment determinant (Fd)-MAK33 with ΔUR variants of λ5 and VpreB in HEK293T cells. This experiment was performed as duplicates. For comparison, the expression and secretion of murine Fd-MAK33 alone and in combination with wild-type λ5, VpreB, and the complete SLC are shown in Supplementary Fig. 3a.

with IgM heavy chains. The $C_H1$ domain of IgM contains a conserved glycosylation site at N46. Since this glycan is crucial for interaction with the λ5-UR, as previously shown (ref. [35]), we compared glycosylated and unglycosylated $C_H1$ (2H32). Surprisingly and in contrast to the IgG $C_H1$ domain, we observed that glycosylated $C_H1$ could not be folded by λ5 but only by the SLC in vitro with a rate constant τ = 66 min, whereas unglycosylated CH1 required τ = 104 min, indicating that glycosylation accelerates $C_H1$ folding by SLC (Supplementary Fig. 3c, d).

Of note, the $K_D$ between $C_L$ and $C_H1$ is 6.2 μM[34] while for $C_H1$ and λ5, the affinity was determined to be slightly higher with a $K_D$ of 2.1 μM (Supplementary Fig. 4a and Table 2). For the complex between $C_H1$ and SLC, the $K_D$ was further decreased to 0.5 μM (Supplementary Fig. 4b and Table 2). This higher affinity seems to be due to an interaction of VpreB with $C_H1$ ($K_D$ 1.3 μM) (Supplementary Fig. 4c and Table 2). This interaction does not induce the folding of $C_H1$ as shown in vitro and in cells. For the interaction of VpreB ΔUR with $C_H1$, the $K_D$ was determined to be 3.3 μM (Supplementary Fig. 4d and Table 2). The interaction of $C_H1$ and VpreB was also verified by AUC, which showed that the two proteins form a complex (Supplementary Fig. 4e). Thus, both the core domain and the UR of VpreB contribute to the $C_H1$ interaction. In the case of λ5, the deletion of the UR does not affect its ability to interact with and fold $C_H1$ (Supplementary Fig. 4f–k and Table 2). In summary, while both SLC proteins interact with $C_H1$, only the core domain of λ5 induces the folding of $C_H1$.

## The unique regions strongly influence the SLC-$V_H$ association

The rearranged $V_H$ domain must be a focus of the SLC-checkpoint. It has been shown previously that VpreB interacts with $V_H$[21,31,32]. However, our understanding of the mechanism, especially the role of the URs, is still at an early stage. To analyze this process further, the interaction of VpreB, λ5 and the SLC with two different $V_H$ domains (murine MAK33 and human 1HEZ) (Supplementary Fig. 5a) was assessed. AUC analyses showed that VpreB and $V_H$ 1HEZ form a heterodimeric complex (Fig. 5a). In this process, VpreB and $V_H$ 1HEZ, which are both homodimers, have to dissociate. Also, our analysis shows that λ5 and $V_H$ form a defined heterodimeric complex (Fig. 5b).

Surprisingly, the affinities between the antibody $V_H$ domain and SLC proteins determined by SPR are remarkably high. For VpreB and $V_H$ 1HEZ, the $K_D$ was 22.2 nM (Fig. 5c and Table 3) and for VpreB and $V_H$ Mak33 the $K_D$ was 10.1 nM (Supplementary Fig. 5b and Table 3). Compared to the isolated VpreB, the SLC binds to the $V_H$s with markedly lower affinity; here the $K_D$ was 129.2 nM for $V_H$ 1HEZ (Fig. 5d and Table 3) and 29.6 nM for $V_H$ MAK33 (Supplementary Fig. 5c and Table 3). λ5 also interacts with the $V_H$ domains albeit with lower affinity, reflected in $K_D$ values of 171.9 nM for $V_H$ 1HEZ (Fig. 5e and Table 3) and 43.1 nM for $V_H$ MAK33 (Supplementary Fig. 5d and Table 3). Taken together, these experiments revealed unexpected features of the interactions of the SLC proteins with $V_H$, which raised the question to what extent the URs are involved.

When we tested the interaction of λ5-ΔUR with $V_H$ 1HEZ, we could not detect a complex (Supplementary Fig. 5e), for both $V_H$ domains tested, binding was observed only with wild-type λ5 and not with λ5-ΔUR, strongly suggesting that the λ5-UR mediates this interaction. The CDR3s of the $V_H$ domains contain two negatively charged and one positively charged residue which may be involved in mediating this interaction. However, for the VpreB-UR the situation is different. First of all, VpreB ΔUR still binds to the $V_H$ domains (Supplementary Fig. 5f, g and Table 3). But, while for wildtype VpreB the $K_D$ values for the two $V_H$ domains were similar, the effects of the UR deletion were strongly influenced by the respective $V_H$ domain: The $K_D$ for $V_H$ 1HEZ was 905.5 nM but it was 31.5 nM for $V_H$ MAK33 (Supplementary Fig. 5f, g and Table 3). Thus, depending on the $V_H$ domain used, the deletion of the VpreB-UR led either to a 3-fold or a 40-fold lower affinity compared to VpreB. In summary, the VpreB-UR is involved in the interaction with $V_H$ and the nature of the $V_H$ domain seems to play an important role concerning the magnitude of the effect. Moreover, the λ5-UR is essential for the interaction of $V_H$ and λ5.

## The λ5-UR is crucial for antigen interaction

Since the SLC-Fd complex resembles a regular antibody Fab fragment, it was interesting whether this Fab-SLC is able to bind antigens and which parts of the SLC contributes to this interaction. To test this, enzyme-linked immunosorbent assay (ELISA) were set up, for which we used the C-terminally FLAG-tagged $V_H$ domain or the Fd fragment of the MAK33 antibody which binds creatine kinase[36]. The Fd fragment was assembled with either the SLC or the LC, respectively, to generate either a SLC-Fd (Fab-SLC) or a LC-Fd (Fab-LC) complex. In the ELISA, the antigen was immobilized, and serial dilutions of the Fab complexes were added. We observed a concentration-dependent interaction between the antigen and both "Fabs" demonstrating that the SLC-Fab complex is capable of antigen-binding. Interestingly, the Fab-SLC showed an affinity for the antigen, which was in the same range as compared to the authentic Fab fragment (8.6 vs. 12.6 nM; Fig. 6a). As expected, the antigen affinities of complexes containing only the $V_H$ domain instead of the Fd fragment were lower than those observed for the "Fab" fragments. For $V_H$-SLC, the $K_D$ for the antigen was 94.1 nM and for $V_H$-$V_L$ it was 155.9 nM (Fig. 6b). Isolated $V_H$ has a $K_D$ of 282.6 nM for the antigen (Fig. 6b).

To dissect which part of the SLC determines the affinity in the $V_H$-SLC complex, the two SLC proteins were screened together with $V_H$. For VpreB-$V_H$, we determined a value of 260.1 nM (Fig. 6b) which is close to the value obtained for $V_H$ alone. Thus, VpreB does not seem to contribute significantly to the antigen interaction because it is unfolded in the absence of λ5. In contrast, λ5-$V_H$ exhibited a $K_D$ of 95.0 nM for the antigen (Fig. 6b) which is identical to the affinity of the SLC-$V_H$ complex.

To further dissect the contributions of the different parts of the SLC proteins to antigen binding, we also tested variants in which the UR was deleted. For VpreB-λ5-ΔUR-$V_H$, the $K_D$ for creatine kinase was 122.0 nM (Fig. 6c, d). The deletion of the VpreB-UR in the VpreB-ΔUR-λ5-$V_H$ complex seems to slightly increase antigen affinity compared to the $V_H$-SLC complex (67.9 vs. 94.1 nM, Fig. 6c, d). This confirms the assumption that the λ5-UR significantly contributes to the antigen interaction. However, the VpreB λ5-ΔUR-$V_H$ exhibited pronounced affinity for the antigen. It is important to note that in the presence of the λ5-ΔUR VpreB is folded. Thus, when the λ5-UR is missing, the ability of VpreB becomes visible. Taken together, these results indicate that in the context of the antigen interaction of the SLC, λ5 is the decisive factor because it establishes the folded state of VpreB and because the λ5-UR interacts strongly with the antigen.

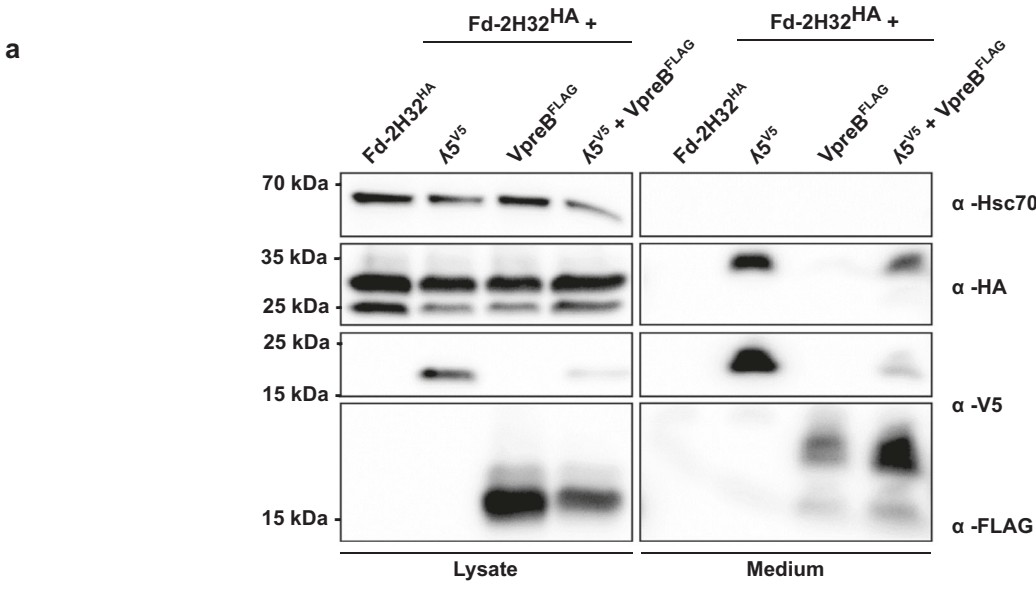

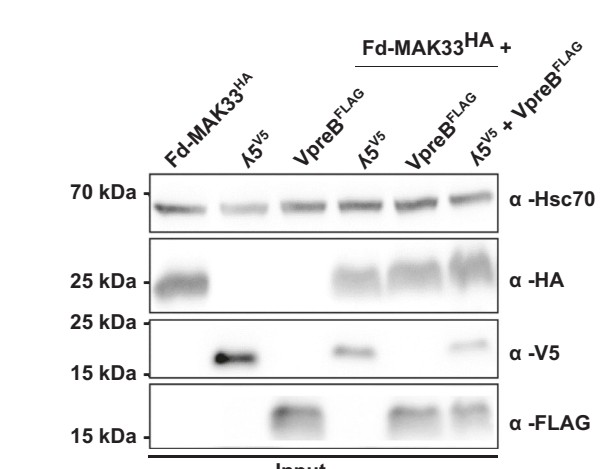

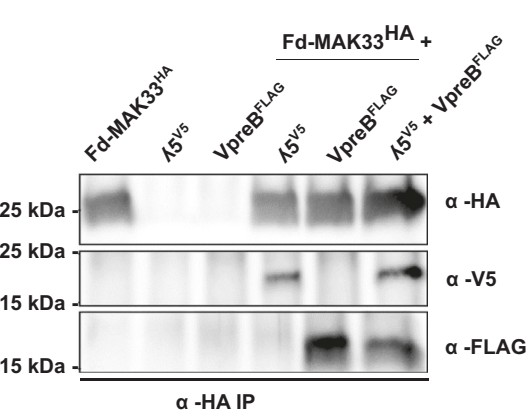

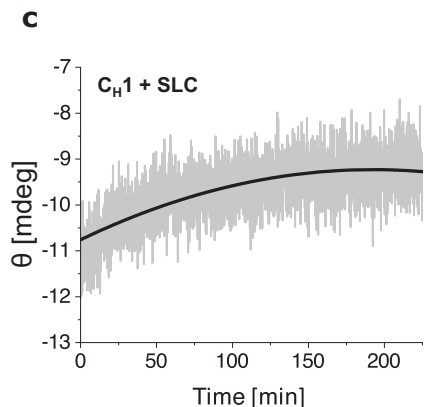

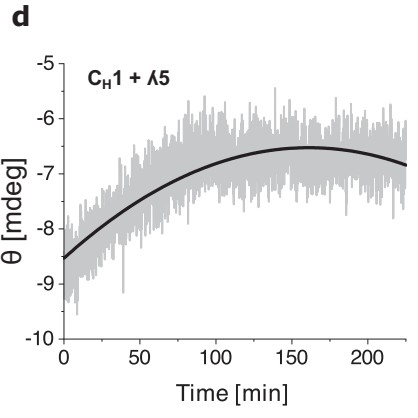

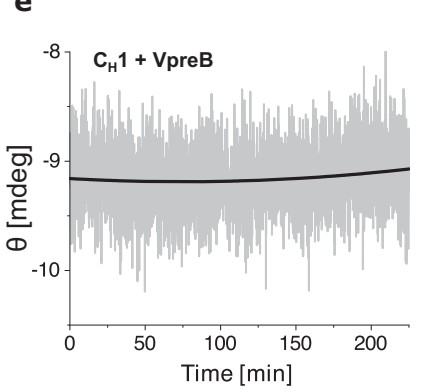

## Discussion

Our study reveals a sophisticated network of assembly steps and conformational transitions that are required to form the SLC hetero-dimer, to govern its interaction with the HC and finally the binding to antigen (Fig. 7). These processes involve interactions in which one of the binding partners VpreB is initially unfolded. VpreB adopts its native structure upon interaction with λ5. This involves a ß-strand located outside the folded domain of λ5, which becomes part of a β-sheet in the folded VpreB structure. This β-strand is necessary and sufficient to induce the folding of the VpreB domain as a peptide corresponding to this strand was able to fold VpreB. Although the in vitro experiments demonstrate that VpreB is unfolded due to the lacking ß6 strand, its

**Fig. 4 | Folding and assembly of the heavy chain (HC) with the surrogate light chain (SLC). a** Expression and secretion of human fragment determinant (Fd)-2H32 alone or together with immunoglobulin lambda 5 (λ5), pre-B lymphocyte protein (VpreB) and the SLC in HEK293T cells. This experiment was performed as duplicate. **b** Immunoprecipitation of murine Fd-MAK33 with ⅄5, VpreB and the SLC in HEK293T lysates. Left: Input samples showing expression of Fd-MAK33-HA, immunoglobulin lambda 5 (λ5)-V5, and VpreB-FLAG, detected by the indicated antibodies. Hsc70 served as a loading control. Right: Immunoprecipitation with an anti-HA antibody to isolate Fd-MAK33-HA, followed by immunoblotting for HA, V5, and FLAG to assess co-immunoprecipitation of λ5 and VpreB. This experiment was performed as duplicate. The folding of constant heavy chain 1 (CH1) MAK33 by SLC (**c**), λ5 (**d**), and VpreB (**e**) was monitored by the change in the circular dichroism (CD) signal at 205 nm. A single exponential trace was observed and fitted by an exponential decay function to obtain the folding rate of the reaction. Representative data from $n = 2$ independent experiments are shown.

### Table 2 | Folding, association kinetics and affinity of constant heavy chain 1 (C_H1) and surrogate light chain (SLC) variants

|            | τ [min]     | $K_D$ [µM]   | $k_{on}$ [1/Ms] | $k_{off}$ [1/s]    |
|------------|-------------|-------------|-----------------|--------------------|
| VpreB      | No folding  | 1.3 ± 0.5   | 2938 ± 1301     | 0.003 ± 0.0003     |
| λ5         | 41.4 ± 2.0  | 2.1 ± 0.1   | 2106 ± 22       | 0.004 ± 0.0002     |
| SLC        | 53.9 ± 4.6  | 0.5 ± 0.3   | 12960 ± 7920    | 0.004 ± 0.00007    |
| VpreB ΔUR  | -           | 3.2 ± 0.1   | 760 ± 31        | 0.002 ± 0.00003    |
| λ5 ΔUR     | 45.0 ± 1.7  | 1.4 ± 0.3   | 2330 ± 232      | 0.003 ± 0.0005     |
| λ5 Δβ      | 43.2 ± 3.7  | 1.1 ± 0.004 | 5010 ± 1740     | 0.005 ± 0.002      |
| λ5 ΔUR Δβ  | 51.6 ± 1.4  | 1.7 ± 0.03  | 2866 ± 87       | 0.005 ± 0.00006    |

The folding time constant τ of C_H1 and SLC variants was determined by far-UV circular dichroism (CD) spectroscopy at 205 nm. The affinity of the complexes was measured by surface plasmon resonance (SPR) to determine the $K_D$ in [nM], $k_a$ in [Ms⁻¹] and the $k_d$ in [s⁻¹]. Given errors indicate the standard deviations.

folding state in the cell may be further affected by molecular chaperones until the interaction with λ5 occurs. The consequence of the VpreB- λ5 interaction is a tight complex with a $K_D$ in the low nanomolar range. Thus, once formed, the SLC heterodimer should remain stably associated. A second association-induced folding event is based on the interaction of the unfolded C_H1 domain of the HC with λ5. Thus, both steps of the assembly processes leading to the SLC-HC complex involve transitions from unfolded to folded domains as a critical step. This allows linking the success of pre-B cell receptor formation to the ER quality control system which is based on detecting unfolded regions in proteins and their retention in the ER, which turns out to be a recurring theme in immunology[37,38].

Both SLC proteins, VpreB and λ5, have unstructured segments protruding at the C- and N-terminus, respectively. They are rightly called unique regions because they do not show sequence homology to other proteins. Their presence seems to be the reason why the SLC has to consist of two genes. Our data show that the URs affect every step in the hierarchical assembly of the SLC-HC complex. Their importance is further highlighted by the finding that their deletion results in a compromised SLC complex formation in cells. As their absence did not have negative consequence on the assembly in vitro, they may have additional functions in the cellular environment such as increasing solubility or shielding regions that can lead to ER retention and/or protein degradation.

Our results reveal a prominent role of all three structural elements of λ5 in the structure formation and function of the SLC and the SLC-HC complex (Fig. 7). Especially, the λ5-UR is of importance as it accelerates the association of the SLC complex, supports the formation of the complex with the HC and is engaged to a large extent in antigen binding. The N-terminal extension of λ5 contains the β-strand sequence required for VpreB folding followed by the 50 residue long UR. During the assembly process with VpreB, this β-strand becomes part of a β-sheet and traverses VpreB from the C-terminal to the N-terminal end[21]. This implies that the λ5-UR protrudes from the VpreB close to the CDR loops and close to the VpreB UR. This localization in the SLC complex puts the λ5-UR in the position to bind to both, the V_H domain and the antigen. We assume that the interaction of the λ5-UR

with V_H keeps the extension in a position required for antigen binding. In this context, it has been reported that the λ5-UR is also involved in the interaction of the pre-B-cell receptor (pre-BCR) with specific ligands[39] and in pre-BCR clustering as well as pre-BCR-mediated signal transduction[29,40–42].

In addition to the importance of the λ5 β-strand and UR, the λ5 core domain exerts an essential function in the quality control process which is required for the pre-BCR to leave the ER and to be transported to the cell surface. Experiments in cells show that the association-induced folding of the intrinsically disordered C_H1 domain is a limiting factor for the ER exit of the SLC and SLC-HC complexes. The in vitro reconstitution of the complexes provided insight into the λ5-induced folding kinetics. The kinetic data we obtained for this association-coupled folding process agree well with those for the C_L-induced folding of the C_H1 domain suggesting that in both scenarios, after forming a partially folded encounter complex, the underlying rate-limiting step for the assembly-induced folding of C_H1 is peptidyl-prolyl isomerization in C_H1[34]. An interesting difference was observed for the glycosylated IgM C_H1-2H32 domain. It folds faster than the unglycosylated form in the presence of the SLC, suggesting that glycosylation enhances the efficiency of C_H1–SLC interactions. This finding is consistent with the in vivo requirement of the conserved C_H1 N-glycosylation for efficient μHC–SLC interaction and pre-BCR surface expression[35]. Notably, we did not observe folding of C_H1-2H32, with or without glycosylation, in the presence of λ5 alone in vitro.

Our results confirm that VpreB alone is present as a largely unfolded oligomer in vitro[30–32]. It could well be that in vivo, the interaction site is affected by the ER-chaperone BiP as binding sites for BiP are present near the interface of VpreB[43]. This interaction could also be responsible for the (partial) ER retention of VpreB observed upon expression in mammalian cells, analogous to the C_H1-BiP interaction[34].

As expected, VpreB binds the V_H domain with high affinity, but we find that it also interacts with the C_H1 domain with moderate affinity. Interestingly, these interactions do not lead to the folding of VpreB or C_H1 but to a dissociation of the VpreB dimer, suggesting that similar interaction sites are involved. These binding events may serve as first proof-reading steps in the scrutiny process of the HC by the SLC. The decreased affinity of VpreB ΔUR for both, the C_H1 domain and even more pronounced the V_H domain, suggests that the UR of VpreB is a determining factor in HC interaction monitoring structural features of the HC. Especially, CDR-H3 seems to be contacted by the VpreB-UR and thus seems to be scrutinized. The interaction scheme revealed for λ5 is even more complex, as it binds VpreB, V_H and C_H1 and antigen. Especially, the binding of the ⅄5-UR to V_H could be a second quality checkpoint for the HC.

Little is known about the interaction of the SLC and the pre-BCR with antigens. We observed a surprisingly high affinity of the Fab-SLC for the antigen of the V_H domain used. The affinity was similar to that of the corresponding regular Fab fragment. Experiments using different SLC variants revealed that the λ5-UR is the determining feature in this interaction as λ5 alone in complex with V_H has a similar affinity for antigen as the SLC with V_H. This effect was abolished when the λ5-UR was deleted. Of note, the λ5-UR also interacts with Galectine-1[39] and the stromal cell ligand ADAM15/fibronectin[44]. Together with our results on antigen interaction, this suggests that the λ5-UR seems to have evolved

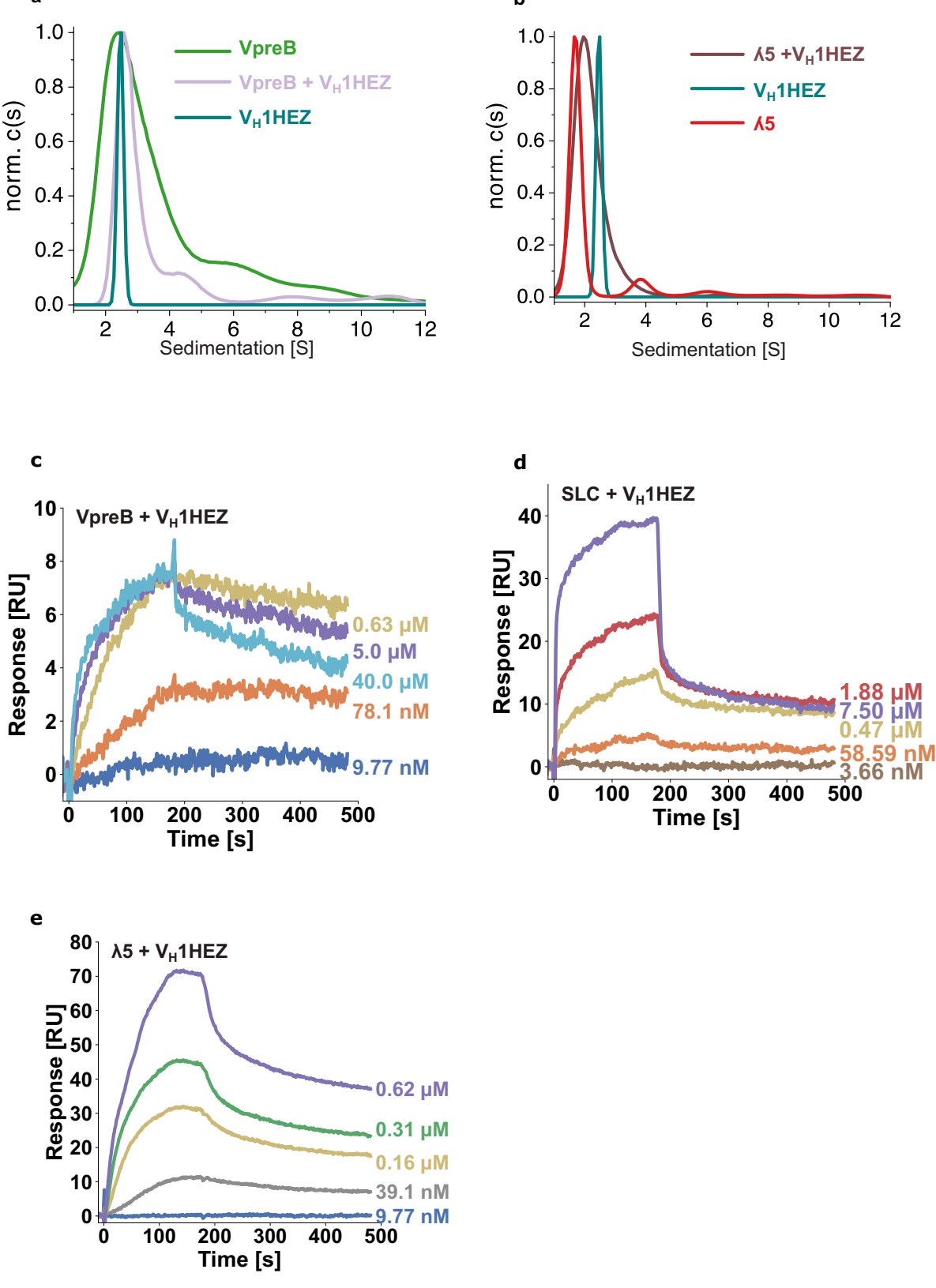

**Fig. 5 | Analysis of the interaction between V_H 1HEZ and V_H MAK33 with SLC.**
The interaction of the V_H 1HEZ and immunoglobulin lambda 5 (λ5) (**a**) and of V_H
1HEZ and pre-B lymphocyte protein (VpreB) (**b**) were determined by analytical
ultracentrifugation and was performed once. Surface plasmon resonance
sensorgrams for the interaction of VpreB with V_H 1HEZ (**c**), of SLC with V_H 1HEZ (**d**),
and of λ5 with V_H 1HEZ (**e**). SPR measurements were performed in $n = 2$ independent
experiments, and representative sensorgrams are shown.

**Table 3 | Analysis of the interaction between $V_H$ domains from the human antibody 1HEZ ($V_H$1HEZ) and the murine antibody MAK33 ($V_H$ MAK33) with SLC variants by surface plasmon resonance (SPR) with the measured constants of $K_D$, $k_{on}$ and $k_{off}$**

| | | $K_D$ [nM] | $k_{on}$ [1/Ms] | $k_{off}$ [1/s] |
|---|---|---|---|---|
| $V_H$ 1HEZ | VpreB | 22.2 ± 9.0 | 41025 ± 10425 | 0.0008 ± 0.0001 |
| | λ5 | 171.9 ± 77.0 | 12606 ± 6584 | 0.002 ± 0.0002 |
| | SLC | 129.2 ± 26.8 | 19745 ± 305 | 0.003 ± 0.0005 |
| | VpreB ΔUR | 905.5 ± 278.5 | 1611 ± 656 | 0.001 ± 0.0001 |
| | λ5 ΔUR | no interaction | - | - |
| | λ5 Δβ | 121.7 ± 27.7 | 32175 ± 3735 | 0.004 ± 0.0004 |
| $V_H$ MAK33 | VpreB | 10.1 ± 3.7 | 65990 ± 25720 | 0.0006 ± 0.00002 |
| | λ5 | 43.1 ± 9.6 | 44945 ± 12655 | 0.002 ± 0.0001 |
| | SLC | 29.6 ± 5.5 | 42745 ± 7155 | 0.001 ± 0.00002 |
| | VpreB ΔUR | 31.5 ± 3.8 | 50185 ± 4105 | 0.002 ± 0.00006 |

The affinity of the complexes was determined by SPR to determine $K_D$, $k_a$, and the $k_d$. Given errors indicate the standard deviations.

as an intrinsically disordered promiscuous binder. This relates this region conceptually to the N-terminal regions of sHsps, which are also unfolded extensions of a folded β-sheet domain and interact with a variety of proteins[45,46].

Our study reveals a complex series of assembly steps that are coupled to protein folding reactions and required to establish the structure of the pre-B cell receptor (Fig. 7). Compared to the assembly of IgG antibodies, formation of the pre-B cell receptor is much more complicated and involves additional structural elements and steps. This may reflect the importance of this checkpoint for successful HC rearrangement during B-cell development and the required connection to the ER quality control system.

## Methods
### Molecular biology and recombinant expression of the SLC proteins and antibody domains
Constructs for λ5 and VpreB variants were synthesized by GeneArt (Regensburg, Germany) and cloned into the pE-SUMOpro vector (Kan$^R$) (Lifesensors, Malvern, USA) after the SUMO Protease 1 cleavage site via the BsaI restriction site using sequence- and ligase-independent cloning[47,48]. Genes for antibody domains were synthesized by Geneart (Regensburg, Germany) and cloned into the pET28b vector (KanR) via the XbaI/Xho restriction sites keeping the sequence between XbaI and NcoI at the 5´ end of the nucleotide sequence of the proteins using sequence and ligase- independent cloning[47,48]. All constructs were sequenced by Eurofins Genomics (Ebersberg, Germany). In all λ5 variants, the C-terminal cysteine (C212), which forms a disulfide bridge with the $C_H$1 domain of the HC in the final assembled state, was mutated to a serine.

Plasmids were transformed into the *E. coli* strain BL21-CodonPlus (DE3)-RIL. Cells were grown at 37 °C in selective LB medium and expression was started by the addition of 1 mM isopropyl-beta-D-thio-galactopyranoside (IPTG) at an OD$_{600}$ of 0.9. After 3 h, cells were harvested (7000×$g$). The harvested cells were resuspended in 50 mM Tris/HCl (pH 7.5), 100 mM NaCl, and lysed using a cell disruptor (Constant Cell Disruption Systems, Northants, UK). A centrifugation step at 12,200×$g$ for 40 min at 4 °C was followed to obtain the inclusion bodies in the pellet.

### Refolding and purification of the SLC proteins, VH MAK33-FLAG, the SLC heterodimer, the Fab and the Fd-SLC complex
Inclusion bodies were solubilized in 100 mM sodium phosphate, pH 7.5, 6 M GdmCl, 10 mM imidazole at 4 °C for 16 h by gentle stirring.

Insoluble components were removed by centrifugation (33,000×$g$, 1 h, 6 °C). The supernatant was applied to a Ni-chelating column (Cytiva, Freiburg, Germany) pre-equilibrated in 50 mM sodium phosphate, pH 7.5, 5 M GdmCl, and 10 mM imidazole (equilibration butter). After washing with five column volumes of equilibration buffer, elution was performed applying a gradient over 10 column up to 500 mM imidazole. For reduction, the purified proteins were incubated for 1 h at RT with 10 mM β-mercaptoethanol and gentle stirring.

Refolding was carried out via drop dilution of the unfolded protein in 100 mM Tris/HCl, pH 8.0, 150 mM NaCl, 350 mM L-arginine, and 10 mM β-mercaptoethanol to a final protein concentration of 0.1 g L$^{-1}$. The diluted protein was then dialyzed against 100 mM Tris/HCl, pH 8.0, 150 mM NaCl, 350 mM L-arginine, 4 mM oxidized glutathione, 0.5 mM reduced glutathione for 36 h at 10 °C and simultaneous addition of His-tagged SUMO protease to remove the N-terminal His-SUMO-tag. A dialysis step against 50 mM HEPES, pH 7.4, 150 mM KCl, 30 mM imidazole for 16 h at 4 °C followed. Aggregates were removed by filtration using a PVDF syringe filter with 0.22-µm pore size.

The filtered protein was applied to a Ni-chelating column, pre-equilibrated in 50 mM HEPES, 150 mM KCl, and pH 7.4 (native buffer). The target protein is in the flowthrough while the SUMO protease remains bound to the column. The flow-through containing the target protein was concentrated using Amicon®Ultra-15 3 K MWCO (Merck KGaA, Darmstadt, Germany) and the concentration was determined from the E280 values determined in a Jasco V-630 Spectrophotometer (Jasco, Grossumstadt, Germany) at 280 nm using the calculated extinction coefficient.

The SLC heterodimer was purified according to the scheme described above with the following changes: The single domains were combined in an equimolar ratio before drop dilution at a concentration of 0.05 g L$^{-1}$ for λ5. For the purification of the Fd-SLC complex, it was similar: The Fd fragment was diluted to a concentration of 0.06 g L$^{-1}$ and equimolar ratios of VpreB and λ5 C212S were added. Fd-LC (Fab complex) was purified as described previously[36].

### Expression and purification of Ulp1-protease (SUMO-protease)
The ubiquitin-like-specific protease 1 (ULP1) (AA 403-621) from *Saccharomyces cerevisiae* containing an N-terminal His$_6$-tag (LifeSensors Inc., Malvern, USA) (pET28b, Kan$^R$) was transformed into the *E. coli* strain BL21-CodonPlus (DE3)-RIL. Cells were grown at 37 °C in selective LB medium, and expression was started by the addition of 1 mM IPTG at an OD$_{600}$ of 0.6–0.8. After 4 h, cells were harvested by centrifugation at 7000×$g$ and 4 °C. The harvested cells were resuspended in 40 mM sodium phosphate, pH 7.4, 300 mM NaCl, 40 mM imidazole (phosphate buffer), and lysed using a cell disruptor (Constant Cell Disruption Systems, Northants, UK). A centrifugation step at 33,000×$g$ and 6 °C for 1 h followed. The supernatant was applied to a pre-equilibrated Ni-chelating column. After washing with phosphate buffer for five column volumes, elution was performed applying a gradient over 15 column volumes up to 300 mM imidazole. The pooled and concentrated protein was applied to a 16/60, 75 pg, Superdex size-exclusion chromatography column (Cytiva, Freiburg, Germany) in 50 mM Tris/HCl, pH 8.0, 500 mM NaCl, 1 mM DTT at 4 °C. Finally, 50% glycerol, and 1% tergitol were added to the purified protein.

### Refolding and purification of antibody domains
Inclusion bodies for antibody domains were solubilized and reduced in 50 mM Tris (pH 7.5), 8 M urea, 10 mM β-mercaptoethanol at 4 °C for 16 h by gentle stirring. Insoluble components were removed by centrifugation (33,000×$g$, 1 h, 6 °C). The supernatant was applied to a Q Sepharose fast flow column (Cytiva, Freiburg, Germany) pre-equilibrated in 50 mM Tris/HCl, pH 7.5, and 5 M urea. After washing for two column volumes, elution was performed applying a gradient over 10 column volumes up to 1 M NaCl. The proteins were diluted to a protein concentration of roughly 0.1 g L$^{-1}$. Refolding of $C_H$1 MAK33 was

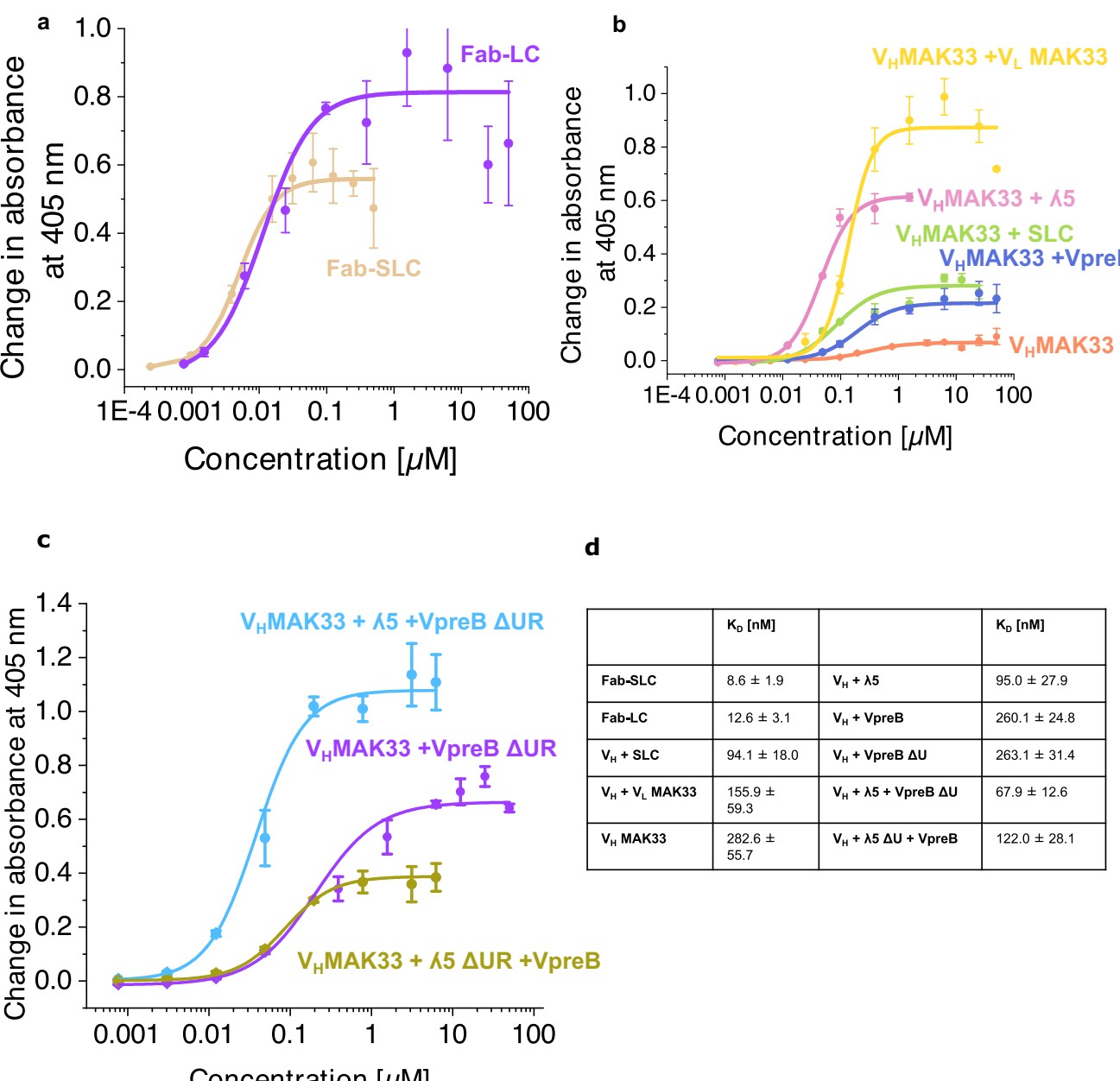

**Fig. 6 | Antigen binding of immunoglobulin lambda 5 (λ5) and pre-B lympho-cyte protein (VpreB) with surrogate light chain (SLC).** Binding to human creatine kinase was determined by enzyme-linked immunosorbent assay (ELISA). **a** fragment antigen-binding (Fab)-light chain (LC). **b** $V_H$ domains in combination with SLC proteins. **c** Influence of the unique regions (UR) on antigen binding. **d** Overview of antigen binding affinity of different SLC constructs. Data were presented as mean values ± SD. from $n = 3$ independent experiments.

carried out via dialysis in 250 mM Tris/HCl, pH 8.0, for 16 h at 10 °C. For $C_L$ MAK33, $V_L$, and $V_H$ 1HEZ, as well as $V_L$ MAK33, dialysis was against 250 mM Tris, pH 8.0, 100 mM L-arginine, 2 mM oxidized glutathione, and 0.5 mM reduced glutathione at 10 °C for 16 h. The proteins were concentrated in Amicon ® Ultra-15 3 K MWCO (Merck KGaA, Darm-stadt, Germany). Aggregates were removed by filtration using a PVDF syringe filter with 0.22-μm pore size. The filtrated protein was applied to a Superdex, 26/60, 75 pg (Cytiva, Freiburg, Germany) size-exclusion chromatography column pre-equilibrated in phosphate-buffered sal-ine (PBS: 137 mM NaCl, 2.7 mM KCl, 10 mM $Na_2HPO_4$, 2 mM $KH_2PO_4$, pH 7.4). After a final concentration step, protein concentrations were determined using their calculated extinction coefficients in a Jasco V-630 Spectrophotometer (Jasco, Grossumstadt, Germany) at 280 nm.

The N-glycosylated form of $C_H1$ was transiently expressed and secreted from Expi293 cells cultured in suspension as $His_6$-Fd,

containing a TEV tag between the $His_6$-$V_H$ and $C_H1$ domains, and co-expressed with a $His_6$-tagged light chain. For protein purification, 200 mL of clarified culture supernatant containing the secreted protein was directly subjected to Ni-NTA affinity chroma-tography. The bound protein was then incubated overnight with TEV protease, and $C_H1$ was eluted the next day using HisTrapA buffer. The protein was further purified by size-exclusion chromatography on a Superdex 75 pg 26/60 column (Cytiva, Freiburg, Germany) pre-equilibrated with phosphate-buffered saline (PBS: 137 mM NaCl, 2.7 mM KCl, 10 mM $Na_2HPO_4$, 2 mM $KH_2PO_4$, pH 7.4) for polishing.

**Peptide**

The additional β-strand of λ5 (λ5β) was obtained as a peptide with the sequence: THVFGSGTQLTVLS from Biomatik USA (Wilmington, USA).

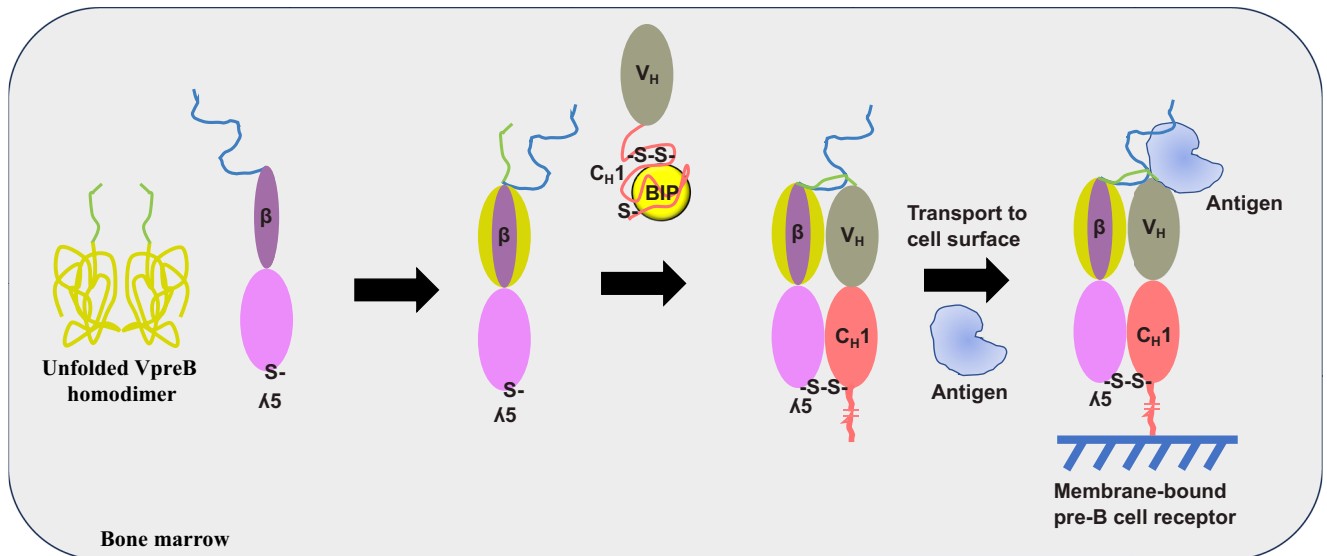

**Fig. 7 | Proposed model for surrogate light chain (SLC) and pre-BCR assembly.** pre-B lymphocyte protein (VpreB) alone is an unfolded homodimer. It is folded upon insertion of the β-strand of immunoglobulin lambda 5 (λ5) creating a high-affinity complex and positioning the two unique regions (UR) at the N-terminal end of VpreB. λ5 also folds the unstructured $C_H1$ domain of the heavy chain (HC) upon binding and relieves the BiP interaction. Furthermore, the λ5 UR is crucial for antigen interaction. The assembled pre-BCR is transported to the cell surface. The figure only shows the two N-terminal domains of the HC.

## Optical spectroscopy

Circular dichroism (CD) spectra were recorded in a Chirascan plus CD spectrophotometer (Applied Photophysics, Leatherhead, UK) to assess the secondary structure. Far-UV CD spectra were recorded in a 1 mm quartz cuvette in PBS buffer (100 mM $Na_2HPO_4 \times 2H_2O$, 18 mM $KH_2PO_4$, 27 mM KCl, 1.37 M NaCl, and pH 7.4). and measured at a protein concentration of 0.12 g $L^{-1}$ between 260 and 200 nm. The scan speed was 0.5 s per point, and spectra were accumulated 16 times, buffer-corrected, and normalized for mean residue molar ellipticity. Thermal transitions were recorded in a 1 mm path length quartz cuvette at 205 nm with a heating rate of 60 °C/h from 20 to 90 °C. Thermal transitions were normalized and fitted using a Boltzmann function to obtain the transition point where half of the protein is unfolded ($T_m$).

Far-UV CD kinetics were recorded in a Jasco J-1500 spectrometer (Jasco, Grossumstadt, Germany) in a 1 mm quartz cuvette at 205 nm and 25 °C for 4 h. The protein concentration was 5 μM per domain in PBS buffer. Thermal transitions of the formed complexes were recorded in a Jasco J-1500 spectrometer (Jasco, Grossumstadt, Germany) equipped with a Peltier element in a 1 mm quartz cuvette at 205 nm with a heating rate of 60 °C/h from 20 to 90 °C. Kinetics were fitted using an exponential decay function to obtain the point where 66.3% of the protein is folded (τ). For each condition, two independent kinetic measurements were performed. The fitting start point was defined as the earliest time after which the signal derivative remained within a stable range, thereby excluding instrumental artefacts.

## Analytical ultracentrifugation (AUC)

AUC measurements were carried out using an Optima AUC I (Beckman, Krefeld, Germany) equipped with absorbance optics. The protein concentration was 20 μM in 50 mM HEPES/KOH, pH 7.4, and 150 mM KCl. About 300 μL sample were loaded into assembled cells with quartz windows and 12-mm-path-length charcoal-filled epon double sector centerpieces. Samples were sedimented at 160,000×g (42,000 rpm) in an eight-hole Beckman-Coulter A50-ti rotor at 20 °C. Sedimentation was continuously scanned with a radial resolution of 30 μm and monitored at 280 nm. Data analysis was carried out with SEDFIT using the continuous c(S) distribution mode[49,50] and the data were normalized.

## Hydrogen-deuterium exchange mass spectrometry (HDX-MS)

HDX was performed on an automated robotic system (HTS PAL; LEAP Technologies, Ft Lauderdale, USA) coupled to an Acquity M-Class UPLC and a HDX manager (Waters Corp., Milford, USA) as described elsewhere (Zhang et al., 2014).

About 3 μL of a 30 μM protein sample was added to 57 μL deuterated buffer (50 mM HEPES/KOH, 150 mM KCl, and pH 7.4) and incubated at 20 °C for 0.17, 1, 10, 30, and 120 min. HDX-MS experiments were performed in two independent experiments, each including three technical replicates per protein and per time point. The exchange was stopped by adding 50 μL of quench buffer (200 mM $Na_2HPO_4$, 200 mM $KH_2PO_4$, 4 M GdmCl, and pH 2.2) to 50 μL of labeled protein at 1 °C. Digestion was performed on-line using an immobilized Waters® Enzymate™ BEH Pepsin Column (2.1 × 30 mm; Waters Corporation, Milford, USA) at 20 °C. Peptides were trapped at 0 °C on a VanGuard Pre-column [Acquity UPLC BEH C18 (1.7 μm, 2.1 × 5 mm, Waters Corporation, Milford, USA)] for 5 min. The peptides were separated using a C18 column [Acquity UPLC BEH C18 (1.0 × 100 mm, Waters Corporation, Milford, USA)] at 0 °C by gradient elution of 0–35% (v/v) acetonitrile (0.1% v/v formic acid) in $H_2O$ (0.1 % v/v formic acid) over 6 min followed by a gradient elution of 35-40% (v/v) acetonitrile (0.1% v/v formic acid) in $H_2O$ (0.1 % v/v formic acid) over 7 min, both gradients at a flow rate of 40 μL $min^{-1}$. Eluting peptides were detected using a Synapt G2S mass spectrometer (Waters Corporation, Milford, USA). The mass spectrometer was operated in $HDMS^E$ mode, with dynamic range enabled (data independent analysis (DIA) coupled with IMS separation) were used to separate peptides prior to collision-induced dissociation (CID) fragmentation in the transfer cell. CID data were used for peptide identification, and uptake quantification was performed at the peptide level (as CID results in deuterium scrambling).

Data were analyzed using Protein Lynx Global Server PLGS (v3.0.3) and DynamX (v3.0.0) software (Waters Corporation, Milford, USA). Search parameters in DynamX were as follows: peptide and fragment tolerances = automatic, min fragment ion matches = 1, digest reagent = non-specific, false discovery rate = 100. Restrictions for peptides in DynamX were as follows: minimum intensity = 5000, minimum products per amino acid = 0, max sequence length = 20, min

sequence length = 5, max ppm error = 0, file threshold = 0. The software Deuteros[51] was used to identify peptides with statistically significant increases/decreases in deuterium uptake based on replicate-derived confidence intervals (99%) and to prepare Wood´s plots. No formal statistical hypothesis testing was performed. An HDX-MS summary table is provided in Supplementary Table 1 in accordance with community reporting standards.

## Constructs for cell experiments

Constructs used in this study for cell culture experiments were obtained from GeneArt (Regensburg, Germany) in a pBUDCE4.1 vector optimized for mammalian expression for SLC constructs. Fd constructs were cloned into the pcDNA 3.1 vector. All constructs were verified by sequencing.

## Cell culture and transient transfections

HEK293T cells (obtained from ECACC) were cultivated in Dulbecco's Modified Eagle's Medium ((DMEM), high glucose (Sigma-Aldrich, Steinheim, Germany), supplemented with 10% (v/v) fetal bovine serum (Biochrom) and 1% (v/v) antibiotic-antimycotic solution (25 µg/ml amphotericin B, 10 mg/ml streptomycin, 10,000 units of penicillin (Sigma-Aldrich, Steinheim, Germany) and GlutaMAX (Thermo Fisher Scientific, Dreieich, Germany) at 37 °C in a 95% humidified atmosphere containing 5% $CO_2$. Transient DNA transfections of HEK293T cells were carried out in p35 dishes (uncoated Nunclon multidish six-well plates) using GeneCellin (BioCellChallenge, Signes, France) according to the manufacturer´s protocol. To perform co-transfections, the total amount of transfected DNA was divided equally according to the manufacturer's protocol.

## Cell lysis

All cell lysis steps were performed on ice using ice-cold solutions. Unless otherwise stated, cell lysis was performed 24 h after transfection. Cells were washed twice with PBS and then 0.5 ml NP-40 lysis buffer (50 mM Tris-HCl, pH 7.5, 150 mM NaCl, 0.5% (w/v) NaDOC, complemented with 0.5% (v/v) NP-40 (Sigma-Aldrich, Steinheim, Germany) and 1x Roche Protease Inhibitor w/o EDTA (Roche Diagnostics, Penzberg, Germany) were added to the cells. After an incubation period of 20 min, the cells were scraped from the plates, and the resulting cell lysate was centrifuged for 15 min at 20,000×g, 4 °C. The supernatant was either used for further analysis or complemented with 5x Laemmli buffer containing 10% (v/v) β-mercaptoethanol. Laemmli-containing sample lysates were subsequently heated for 10 min at 95 °C.

## Deglycosylation experiments

Deglycosylation assays with a mix of O-glycosidase and α2-3,6,8 Neuraminidase (NEB, Frankfurt, Germany) were performed according to the manufacturer's instructions at 37 °C for 1 h. The digested sample proteins were thereafter supplemented with 5x Laemmli buffer and 10% (v/v) β-mercaptoethanol and heated for 10 min at 95 °C.

## Secretion experiments

For secretion assays, cells were transfected for 24 h. The cells were then washed twice with PBS (Sigma-Aldrich, Steinheim, Germany) and subsequently supplemented with 0.5 ml fresh DMEM for cultivation for another 24 h. To analyze secreted proteins, the medium was removed from the cells and centrifuged for 5 min at 300×g and 4 °C. The resulting supernatant was thereafter transferred into a new reaction tube and supplemented with 0.1 volumes of 500 mM Tris/HCl (pH 7.5), 1.5 M NaCl, complemented with 10x Roche Complete Protease Inhibitor w/o EDTA (Roche Diagnostics, Penzberg, Germany). After another centrifugation step for 15 min at 20,000×g and 4 °C, 5x Laemmli buffer containing 10% β-mercaptoethanol (β-Me) was added to the samples. Samples were then heated for 10 min at 95 °C. The

actual cell lysis was then carried out using a PBS wash step before the addition of 0.5 ml of 1x RIPA lysis buffer (50 mM Tris/HCl, pH 7.5, 150 mM NaCl, 1% NP-40, 0.5% DOC, and 0.1% SDS) supplemented with 1x protease inhibitor, to the cells. After 20 min, the cell lysate was centrifuged for 15 min at 20,000×g and 4 °C and 0.2 volumes of 5x Laemmli buffer containing 10% β-mercaptoethanol (β-Me) were added. Samples were thereafter heated to 95 °C for 10 min.

## Co-immunoprecipitation

Protein–protein interactions were analyzed and detected using co-immunoprecipitations (co-IPs). Before co-IP of target proteins, cells were lysed using NP-40 lysis buffer, and 2% of the cell lysate was supplemented with 5x Laemmli buffer containing 2% (v/v) β-mercaptoethanol. The remaining lysate was then incubated with 1.5 µg of mouse monoclonal anti-HA antibody for 2 h under constant rotation at 4 °C. Following this, 30 µl protein A/G agarose beads (Thermo Fischer, Dreieich, Germany) were added and samples incubated for another hour under constant rotation at 4 °C. Thereafter, the beads were washed three times with 1 ml NP-40 wash buffer (50 mM Tris/HCl, pH 7.5, 400 mM NaCl, 0.5 % (w/v) NaDOC, and 0.5 % (v/v) NP-40), including centrifugation steps for 5 min at 4 °C and 620×g (2.500 rpm) in between. Finally, the proteins were eluted by adding 2x Laemmli buffer supplemented with 10% (v/v) β-mercaptoethanol and heating at 95 °C for 10 min.

## SDS-PAGE and immunoblotting

For immunoblots, samples were separated on 15% SDS-PAGE gels and then wet-transferred overnight at 4 °C. After transferring the proteins to methanol (Sigma-Aldrich), activated PVDF membranes (Biorad), the membranes were blocked at least 3 h at RT with Tris-buffered saline supplemented with skim milk powder and Tween-20 (M-TBST; 25 mM Tris/HCl, pH 7.5, 150 mM NaCl, 5% (w/v) skim milk powder, 0.05% (v/v) Tween-20). Primary antibodies were diluted in M-TBST and applied to the membranes overnight at 4 °C. After washing of the membranes (1 × 5 min TBS, 2 × 5 min TBST, 3 × 5 min TBS), HRP-conjugated secondary antibodies diluted in M-TBST were applied to the blots. Following a second washing procedure, proteins were then detected using Amersham ECL prime solution (Cytiva, Freiburg, Germany) on a Fusion Pulse 6 imaging system (Vilber Lourmat, Eberhardzell, Germany).

## Antibodies

For western blots, the following primary antibodies were used at the dilutions listed: rabbit polyclonal anti-FLAG (F7425, Sigma-Aldrich, Steinheim, Germany) at 1:1000; rabbit polyclonal (9023) anti-HA (902302, Biolegend, Koblenz, Germany) at 1:1000; mouse monoclonal (B-6) anti-Hsc70 (sc-7298, Santa Cruz, Heidelberg, Germany) at 1:1000 and mouse monoclonal anti-V5 (680602, Biolegend, Koblenz, Germany) at 1:1000. The following HRP-conjugated secondary antibodies were used for development of Western blots at 1:10,000: mouse IgGκ-binding protein (m-IgGκ BP-HRP) (sc-516102, Santa Cruz, Heidelberg, Germany) and mouse monoclonal anti-rabbit IgG-HRP (sc-2357, Santa Cruz, Heidelberg, Germany). For IP, the following antibodies were employed: monoclonal mouse anti-HA (16B12) (901518, BioLegend, Koblenz, Germany). Uncropped scans of representative blots are provided in the Source Data file.

## Enzyme-linked immunosorbent assay (ELISA)

Pierce™ Streptavidin Coated Clear 96-Well plates (Thermo Fisher Scientific, Dreieich, Germany) were incubated with 90 µL of antigen solution per well at 25 °C for 45 min, shaking on a Thermomixer compact (Eppendorf, Hamburg, Germany) at 350 RPM. This serves for immobilization of the antigen, which is human biotinylated muscle-type creatine kinase. After discarding the antigen solution, each well was washed three times with 150 µL of de-ionized $H_2O$.

Next, a serial dilution of the constructs to be analyzed in a stoichiometric ratio of 1:1 with $V_H$ MAK33-FLAG was prepared in triplicates, except for Fd-SLC and Fd-LC, for which the serial dilutions were prepared directly from the purified complexes. About 50 μL of each protein concentration of the serial dilution was added to the 96-well plate and again incubated at 25 °C for 45 min, shaking at 350 RPM. Another washing step of three times with 150 μL of de-ionized $H_2O$ followed. Thereafter, to each well 100 μL of an anti-FLAG antibody conjugated to horseradish peroxidase (HRP) (Sigma-Aldrich, Steinheim, Germany) in a dilution of 1:15,000 were added, and the plate was incubated at 25 °C for 45 min, shaking at 350 RPM, covered from light. After discarding the solution, each well was washed three times with 150 μL of de-ionized $H_2O$.

Finally, 100 μL of ABTS solution was added per well, and the plate was covered with a sealing foil. Enzymatic conversion of ABTS by HRP leads to an increase in absorption at a wavelength of 405 nm. The absorption was measured in an Infinite M Nano plate reader (Tecan Group, Männedorf, Switzerland) at 405 nm and 25 °C. The measurement was conducted for 1 h with data intervals of 5 min. The intensities after 1 h were taken for data analysis. The absorption was plotted against the concentration on a logarithmic scale and fitted using a Hill1 function to obtain the affinity ($K_D$) for the antigen.

### Surface plasmon resonance (SPR) analysis

The interaction analysis between λ5 and VpreB variants, $C_H1$ and SLC variants, as well as between the $V_H$ domain and SLC variants, was performed with a Biacore X100 instrument (GE Healthcare, Freiburg, Germany) in 50 mM HEPES/KOH, pH 7.4, 150 mM KCl, 3 mM EDTA, and 0.5% Tween-20 as running buffer at 25 °C. The ligands were immobilized on a CM5 sensor chip by amine coupling chemistry at pH 4–6 (Biacore manual, Cytiva Sweden AB, Uppsala, Sweden) using the amine coupling kit (Cytiva Sweden AB, Uppsala, Sweden). Multi-cycle runs with titrations of the analytes at various concentrations were measured with an association time of 3 min, a dissociation time of 5 min and a constant flow rate of 30 μL/min. The sensor surface was regenerated between each run with a 30 s injection of 1.5 M KCl at a flow rate of 30 μL/min.

For the SPR experiments monitoring binding between λ5 and VpreB variants, VpreB was immobilized on a CM5 sensor chip, and purified λ5 was injected as analyte at concentrations ranging from 0.002 to 4 μM. For the SPR experiments between $C_H1$ and SLC variants, $C_H1$ was immobilized on a CM5 sensor chip, and purified SLC variants were injected as analytes at concentrations ranging from 0.002 to 50 μM, depending on the variant used. For the SPR experiments monitoring binding between the $V_H$ domain and SLC variants, $V_H$ 1HEZ or $V_H$ MAK33 was immobilized on a CM5 sensor chip via amine coupling, and purified VpreB, λ5, or the SLC were injected as analytes at concentrations ranging from 0.003 to 10 μM, depending on the variant used.

Binding curves were corrected by subtraction of buffer and reference flow cell signals (binding to control flow cells and running buffer) and fitted to a 1:1 binding model. Sensorgrams were analyzed using the Biacore X100 Evaluation Software (Cytiva Sweden AB, Uppsala, Sweden) to obtain the equilibrium dissociation constant ($K_D$), association rate constant ($k_a$), and dissociation rate constant ($k_d$).

### Nuclear magnetic resonance (NMR) spectroscopy

All spectra were acquired at 25 °C using a Bruker AVANCE600 (Bruker, Rheinstetten, Germany). All proteins were solubilized in 50 mM HEPES, 150 mM KCl, pH 7.4, supplemented with 10% D2O employing a 5-mm-NMR tube. For the measurement of VpreB in the presence of the β-strand derived peptide λ5β, a 150 μM solution of $^{15}$N-labeled VpreB was titrated with unlabeled λ5β to yield a twofold excess of λ5-β. Prior to the experiment, the VpreB-λ5β complex was incubated for at least 2 h at RT to ensure complete folding of VpreB.

Complexes of VpreB and ƛ5 were obtained by co-purification of $^{13}$C,$^{15}$N-labeled VpreB and unlabeled λ5β. For references, the individual proteins ($^{15}$N-labeled VpreB and $^{13}$C,$^{15}$N-labeled λ5β) were measured independently.

To probe the structure of VpreB in the presence or absence of folding agents such as λ5β or full-length λ5, $^{15}$N-HSQC spectra were recorded of VpreB alone at 25 °C by using selective proton flip-back techniques for fast pulsing. Processing was identical for all spectra was performed in TOPSPIN 4.0.3 (Bruker Biospin). The data was further analyzed and visualized in CCPN V3.0.4 (University of Leicester, United Kingdom). The figures were created in Affinity Designer (Serif Europe, Nottingham, UK).

### Statistics and reproducibility

No statistical method was used to predetermine sample size. No data was excluded from the analyses. The experiments were not randomized, and the investigators were not blinded to allocation during experiments and outcome assessment.

Data were presented as mean values, with error bars representing the standard deviation (SD), as indicated in the figure legends. The number of independent experiments is specified in the figure legends. Unless otherwise stated, all experiments were independently repeated at least two times with similar results.

### Reporting summary

Further information on research design is available in the Nature Portfolio Reporting Summary linked to this article.

## Data availability

All data were contained within the main article and Supplementary Information. The source data behind all figures and Supplementary Figs. are included in the source data. Further information can be obtained from the authors upon reasonable request. All data were analyzed using commercially available software as described in the Methods section. Custom scripts, if any, were used only for figure generation and not for data processing or statistical analysis.2H32 [https://www.rcsb.org/structure/2H32]. 1HEZ. MAK33 [https://www.rcsb.org/structure/1FH5]. Source data are provided with this paper.

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

## Acknowledgements
This work was supported by grants from the Deutsche Forschungsgemeinschaft to J.B. (BU 836/6-3) and M.J.F. (B11, SFB 1035). We thank Liane Gretschel and Daniel Weinfurtner (Roche Diagnostics, Penzberg) for providing CK-MM.

## Author contributions
Experiments were designed and performed by J.K., N.C.S.A., N.B., O.S., F.R., and M.R. The manuscript was written by J.B., J.K., M.J.F., R.H., and B.R. Figures were prepared by J.K., R.H., O.S., B.R., M.J.F., and J.B.

## Funding

## Competing interests
The authors declare no competing interests.
