## [Transparent Peer Review file · Nature Communications]

Association-induced folding governs surrogate light chain and pre-B cell receptor core assembly

Corresponding Author: Dr Johannes Buchner

Version 0:

Reviewer comments:

Reviewer #1

(Remarks to the Author)

In the manuscript NCOMMS-23-62879 Koenig et al. study the folding and assembly of components of the surrogate light-chain (VpreB and $\lambda 5$) of the pre-B cell receptor (pre-BCR).

The surrogate light-chain (SLC) forms together with a membrane-bound immunoglobulin (Ig) heavy chain of class M (mIgM) and the CD79a/CD79b signaling components the pre-BCR that promotes the further development of only those pre-B cells with a productive heavy chain (HC) gene assembly. The authors find that in the absence of a HC fragment (Fd), the CL domain of $\lambda 5$ is properly folded while the V domain of VpreB is unfolded. This is a well-known fact as VpreB is lacking one β -strand, that is provided by the N-terminal extension of the $\lambda 5$ protein (ref 21). Interestingly, VpreB and $\lambda 5$ together are forming a SLC in the absence of a Fd HC fragment. The authors also show that similar to a conventional LC, $\lambda 5$ can promote the proper folding of the unstructured CH1 domain. By generating deletion mutants, the authors study the function of the unstructured unique region (UR) of VpreB and $\lambda 5$. They found that the UR of VpreB and $\lambda 5$, promote and reduced the speed of SLC formation.

The Ig domain folding studies are well conducted and described. However, this manuscript has a major problem namely that the authors are not aware of or are ignoring major biological features of the pre-BCR. A functional pre-BCR can only be formed by the first HC class expressed namely mIgM (Uebelhart et al. (2010) Nat Immunol 11: 759-765). The reason for this is that only the CH1 domain of mIgM (but not of mIgD or other mIg classes) carries a conserved asparagine (N46) that is glycosylated and that via the negatively charged sugar moiety apparently can interact with the positive charged UR of the $\lambda 5$ protein. This UR contains several evolutionary highly conserved arginine residues (not at all mentioned in the manuscript). It is thought that that via these highly conserved arginine the UR of $\lambda 5$ binds to the CH1 domain of mIgM and that this releases the inhibitory role of the $\lambda 5$ UR for pre-BCR assembly that the authors also find in their studies. In addition, the N46/UR of $\lambda 5$ interaction can promote the active (constitutive signaling) conformation of the pre-BCR. In their manuscript the authors do not mention the class of the used Fd HC fragment and thus it is questionable whether they study the formation of the biological relevant pre-BCR. Given this major problem I think that the manuscript is more suitable for a biochemical Journal than for Nature Communications.

Specific comments

The sentence on page 3 "The assembly of the SLC with the HC forms the pre-B cell receptor" is wrong as a functional pre-B cell receptor also contains the CD79a/CD79b signaling components

The sentence on page 3 "The non-covalent association with the HC is supported by an intermolecular disulfide bond between $\lambda 5$ and the CH1 domain" is misleading as the intermolecular disulfide bond between $\lambda 5$ and the CH1 domain establishes a covalent association between the SLC and the HC

On page 4 a ref is missing "In the available crystal structure of the pre BCR, the URs are only partially resolved (REF)"

The sentence on page 5 "This is surprising as VpreB which corresponds to the VL domain of a regular LC and isolated VL domains have been shown to be folded in the preB cell receptor" is misleading as the author later explain that the unassembled VpreB component is lacking a β -strand and thus does not correspond to a VL domain.

On page 6 the authors write "Interestingly, the sedimentation coefficient of VpreB alone was ~ 3.2 S, which is indicative of a non-native homodimer, as described previously". How can an unfolded protein like VpreB lacking a β -strand form a homodimer? Have the authors done these sedimentation experiment also with the VpreB that contains the β -strand peptide?

On page 6 the authors write "that both, $\lambda 5$ and VpreB, can interact individually with Fd". Are these binding experiments done with the unfolded VpreB or the VpreB that contains the β -strand peptide?

On page 6 the authors write our result "strongly suggesting that binding of $\lambda 5$ to VH is mediated exclusively by the $\lambda 5$ UR". Is this interaction shown by all the VH domains tested and does the VH domain carry a negatively charged CDR3 region or a

sugar moiety that would explain this otherwise puzzling interaction?

The section on "The $\lambda 5$ -UR is crucial for antigen interaction" is lacking a specificity control. Is the creatine kinase binding only seen by the VH MAK33 or also VH1HEZ containing SLC complexes?

Reviewer #2

(Remarks to the Author)

In this manuscript, König et al. examined the assembly of the pre-B cell receptor in vitro using purified proteins and in HEK293 cells by transient transfection. The pre-B cell receptor, whose crystal structure was reported in 2007, comprises a surrogate light chain (SLC) covalently paired with an antibody heavy chain. The SLC consists of two non-covalently associated proteins, VpreB and $\lambda 5$. The authors show that VpreB, which is unfolded in isolation, adopts its native structure upon interaction with $\lambda 5$. This process (β -strand complementation) requires a β -strand from I5 to provide the missing ninth β -strand of VpreB. In addition, folding of CH1 requires interaction with $\lambda 5$. The biochemical analysis of the assembly process is technically well done and the results largely justify the authors' conclusions. However, it is not quite accurate to claim that "little is known about the assembly process of VpreB and $\lambda 5$ " (p. 4, lines 4-5). In fact, quite a lot was already known, as evident from the references cited. For example, β -strand complementation by $\lambda 5$ was described in refs. 29 and 30. Nevertheless, the present work provides the most complete picture to date of pre-B cell receptor assembly. Points to address:

p. 5, 3rd paragraph. Individual SLC proteins for biophysical analysis were produced by in vitro refolding from inclusion bodies. CD and NMR analyses indicate that isolated VpreB is unfolded. However, finding appropriate conditions for in vitro folding of proteins is often not straightforward. That is to say, failure to refold a protein in vitro does not necessarily mean that the protein is unfolded when produced naturally within a cell in an environment that includes chaperones, etc. The authors should include this caveat. Having said that, it is certainly striking that addition of $\lambda 5$ peptide to isolated VpreB induces folding of VpreB, as indicated by NMR.

Fig. 1c. Figure legend should include Hsc70 and explain that it is a positive control. In medium, there are two bands for VpreB in the $\lambda 5 + VpreB$ lane. What is the lower MW band, which does not appear (or just barely) in the VpreB alone lane? Is it another glycoform of VpreB?

Fig. 2d. Figure legend is incomplete. How exactly were the SPR measurements done? What was immobilized? What was the analyte? Are the concentrations for the $\lambda 5\beta$ peptide? The description of SPR in Materials and Methods does not answer these specific questions.

p. 6, 3rd paragraph. In AUC, isolated VpreB appears to sediment as a homodimer. How is this possible if isolated VpreB is unfolded? In looking at the sedimentation profile of isolated VpreB in Fig. 2f, the profile is very broad compared to those of the other samples, which we know are folded. Very likely the broad VpreB profile reflects non-specific aggregation rather than discrete dimerization. The authors should revise their text accordingly.

Fig. 3g. Figure legend should explain labeling of samples. Presumably, $\lambda 5DU$ means $\lambda 5$ with URL deleted and VpreBDU means VpreB with UR deleted. Also, for comparison of intensities, the immunoblots should include $\lambda 5$ and VpreB with URs intact.

Fig. 4b. Figure legend does not adequately describe immunoprecipitations in the two panels.

Fig. 5c-e. Same comments about SPR as above for Fig. 2d.

p. 9, 3rd paragraph. Results of ELISA assays should be confirmed by SPR using immobilized creatine kinase. The authors have all necessary reagents to make these measurements.

Minor:

On p. 2, 1st line, (REF) should give reference number.

In Fig. 3 legend, panels b and e should be panels b and d.

Version 1:

Reviewer comments:

Reviewer #1

(Remarks to the Author)

The authors have now familiarised them with and cite the existing literature concerning the pre-BCR structure and function.

They also have now addressed most of my comments to improve their MS

Reviewer #2

(Remarks to the Author)

The authors have responded satisfactorily to the previous critiques.

POINT TO POINT REPLY TO THE REVIEWER COMMENTS

Reviewer #1

In the manuscript NCOMMS-23-62879 Koenig et al. study the folding and assembly of components of the surrogate light-chain (VpreB and $\lambda 5$) of the pre-B cell receptor (pre-BCR).

The surrogate light-chain (SLC) forms together with a membrane-bound immunoglobulin (Ig) heavy chain of class M (mIgM) and the CD79a/CD79b signaling components the pre-BCR that promotes the further development of only those pre-B cells with a productive heavy chain (HC) gene assembly. The authors find that in the absence of a HC fragment (Fd), the CL domain of $\lambda 5$ is properly folded while the V domain of VpreB is unfolded. This is a well-known fact as VpreB is lacking one β -strand, that is provided by the N-terminal extension of the $\lambda 5$ protein (ref 21). Interestingly, VpreB and $\lambda 5$ together are forming a SLC in the absence of a Fd HC fragment. The authors also show that similar to a conventional LC, $\lambda 5$ can promote the proper folding of the unstructured CH1 domain. By generating deletion mutants, the authors study the function of the unstructured unique region (UR) of VpreB and $\lambda 5$. They found that the UR of VpreB and $\lambda 5$, promote and reduced the speed of SLC formation. The Ig domain folding studies are well conducted and described. However, this manuscript has a major problem namely that the authors are not aware of or are ignoring major biological features of the pre-BCR. A functional pre-BCR can only be formed by the first HC class expressed namely mIgM (Uebelhart et al. (2010) Nat Immunol 11: 759-765). The reason for this is that only the CH1 domain of mIgM (but not of mIgD or other mIg classes) carries a conserved asparagine (N46) that is glycosylated and that via the negatively charged sugar moiety apparently can interact with the positive charged UR of the $\lambda 5$ protein. This UR contains several evolutionary highly conserved arginine residues (not at all mentioned in the manuscript). It is thought that that via these highly conserved arginine the UR of $\lambda 5$ binds to the CH1 domain of mIgM and that this releases the inhibitory role of the $\lambda 5$ UR for pre-BCR assembly that the authors also find in their studies. In addition, the N46/UR of $\lambda 5$ interaction can promote the active (constitutive signaling) conformation of the pre-BCR. In their manuscript the authors do not mention the class of the used Fd HC fragment and thus it is questionable whether they study the formation of the biological relevant pre-BCR. Given this major problem I think that the manuscript is more suitable for a biochemical Journal than for Nature Communications.

We thank the reviewer for pointing out the importance of the glycosylated mIgM C_{H1} domain for pre-BCR formation. Indeed, this feature had escaped our attention. In the revised manuscript we now performed the first comparative studies of glycosylated and unglycosylated C_{H1} domains, extending the scope of our study. In detail, in response to the reviewer's comment, we established the expression and purification of this glycosylated C_{H1} domain over the past months and performed folding studies with the SLC proteins and the assembled SLC complex.

As expected by the reviewer, using the glycosylated C_{H1} made a difference concerning the interaction with the SLC. Folding of the glycosylated C_{H1} domain by the SLC was faster.

Interestingly, while $\lambda 5$ was sufficient to fold the unglycosylated C_{H1} domain, for the glycosylated version, the entire SLC complex was required. We incorporated these exciting results into the revised version and changed our model accordingly (p. 8): "The MAK33 C_{H1} used in our experiments represents an IgG C_{H1} domain. The pre-BCR is assembled with IgM heavy chains. The C_{H1} domain of IgM contains a conserved glycosylation site at N46. Since this glycan is crucial for interaction with the $\lambda 5$ UR, as previously shown (Uebelhart et al.³⁵), we compared glycosylated and unglycosylated C_{H1} (2H32). Surprisingly and in contrast to the IgG C_{H1} domain, we observed that glycosylated C_{H1} could not be folded by $\lambda 5$ but only by the SLC in vitro with a rate constant $\tau = 66$ min, whereas unglycosylated CH1 required $\tau = 104$ min, indicating that glycosylation accelerates C_{H1} folding by SLC (Supplementary Fig. 3c-d)."

And in discussion (p. 12): "An interesting difference was observed for the glycosylated IgM CH1-2H32 domain. It folds faster than the unglycosylated form in the presence of the SLC, suggesting that glycosylation enhances the efficiency of CH1–SLC interactions. This finding is consistent with the in vivo requirement of the conserved CH1 N-glycosylation for efficient μ HC–SLC interaction and pre-BCR surface expression³⁵. Notably, we did not observe folding of CH1-2H32, with or without glycosylation, in the presence of $\lambda 5$ alone in vitro."

Specific comments

The sentence on page 3 “The assembly of the SLC with the HC forms the pre-B cell receptor“ is wrong as a functional pre-B cell receptor also contains the CD79a/CD79b signaling components

We agree that our original wording was not precise, and we have revised the sentence accordingly.

Revised text (p.3):

“The assembly of the SLC and the HC together with the CD79a/CD79b signaling components forms the pre-B cell receptor ¹⁰⁻¹²“

The sentence on page 3 “The non-covalent association with the HC is supported by an intermolecular disulfide bond between $\lambda 5$ and the CH1 domain“ is misleading as the intermolecular disulfide bond between $\lambda 5$ and the CH1 domain establishes a covalent association between the SLC and the HC

We have revised the sentence to clarify the distinction between non-covalent interactions and the disulfide bond. The revised text (p. 3) now reads:

“The association of the SLC with the HC involves both non-covalent interactions and a covalent intermolecular disulfide bond between $\lambda 5$ and the CH1 domain ^{3, 24-28}“

On page 4 a ref is missing “In the available crystal structure of the pre BCR, the URs are only partially resolved (REF)“

Response: The reference has been added, apologies for that.

The sentence on page 5” This is surprising as VpreB which corresponds to the VL domain of a regular LC and isolated VL domains have been shown to be folded in the preB cell receptor“ is misleading as the author later explain that the unassembled VpreB component is lacking a β -strand and thus does not correspond to a VL domain.

We thank the reviewer for highlighting this inconsistency. We agree that while VpreB is homologous to a VL domain, it is not equivalent, as it lacks a conserved β -strand and therefore does not correspond to a complete VL domain. We have revised the sentence to clarify this distinction. The revised text (p. 5) now reads:

“Although VpreB shares homology with the VL domain of a conventional light chain, it lacks a conserved β -strand and therefore cannot fold independently into a stable Ig-like domain. Importantly, conventional VL domains cannot replace VpreB in the pre-BCR, showing that VpreB and $\lambda 5$ together provide unique structural features required for pre-BCR assembly.”

On page 6 the authors write ”Interestingly, the sedimentation coefficient of VpreB alone was ~ 3.2 S, which is indicative of a non-native homodimer, as described previously“. How can an unfolded protein like VpreB lacking a β -strand form a homodimer? Have the authors done these sedimentation experiment also with the VpreB that contains the β -strand peptide?

We thank the reviewer for this comment. One would expect that an unfolded protein should not have any defined structure and therefore would be unable to oligomerize. Findings reported in the literature, however, show that assembly is compatible with disorder

(<https://www.nature.com/articles/nature25762>). Of note, we observed a broad peak at approximately 3.2 S, which roughly corresponds to the dimer–tetramer range of VpreB. Because the peak is very broad rather than sharp, it likely reflects a heterogeneous mixture of oligomeric states of VpreB. An aggregate would be much larger and therefore would not be visible in this sedimentation region.

Regarding the suggested experiment with the β -strand peptide, we did not perform AUC measurements with VpreB in the presence of this peptide due to limited availability. However, our NMR data (Fig. 1e) clearly demonstrate that addition of the $\lambda 5\beta$ peptide induces folding of VpreB, supporting the interpretation that $\lambda 5$ contributes the missing β -strand required for proper folding.

We revised the manuscript to address this point more clearly (p. 6):

“For isolated VpreB, the sedimentation coefficient was also ~ 3.2 S; however, the broad profile compared to folded controls suggests that this signal arises from a heterogeneous mixture of oligomeric species in the range of dimers to tetramers. This interpretation is consistent with previous

reports describing non-native dimeric or higher-order VpreB assemblies formed under certain in vitro conditions³⁰⁻³².”

On page 6 the authors write” that both, $\lambda 5$ and VpreB, can interact individually with Fd“. Are these binding experiments done with the unfolded VpreB or the VpreB that contains the β -strand peptide? The binding experiments shown in Fig. 4a were performed in HEK293T cells with the wild-type VpreB construct, which does not contain the additional β -strand peptide. We have revised the sentence on page 6 to clarify this point(p. 8):

“...that both $\lambda 5$ and VpreB, when expressed in HEK293T cells, can interact individually with Fd. Thus, for VpreB, the lacking β -strand is not a prerequisite for Fd binding.”

On page 6 the authors write our result” strongly suggesting that binding of $\lambda 5$ to V_H is mediated exclusively by the $\lambda 5$ UR“. Is this interaction shown by all the V_H domains tested and does the V_H domain carry a negatively charged CDR3 region or a sugar moiety that would explain this otherwise puzzling interaction?

We thank the reviewer for this important question. We tested two V_H domains in combination with $\lambda 5$ and $\lambda 5$ - Δ UR. In both cases, complex formation was observed with wild-type $\lambda 5$ but not with $\lambda 5$ - Δ UR, indicating that the $\lambda 5$ UR is required for V_H binding. The CDR3 of the V_H domains tested contains two negatively charged and one positively charged residue, but no N-linked glycosylation motif. Thus, the observed $\lambda 5$ -V_H interaction cannot be attributed to CDR3 glycosylation. We have revised the manuscript text to clarify these points(p. 9):.

“...for both V_H domains tested, binding was observed only with wild-type $\lambda 5$ and not with $\lambda 5$ - Δ UR, strongly suggesting that the $\lambda 5$ UR mediates this interaction. The CDR3s of the V_H domains contain two negatively charged and one positively charged residue which may be involved in mediating this interaction”

The section on “The $\lambda 5$ -UR is crucial for antigen interaction“ is lacking a specificity control. Is the creatine kinase binding only seen by the V_H MAK33 or also V_H1HEZ containing SLC complexes?

We appreciate this comment. In the present study, antigen binding to human creatine kinase (muscle type) was assessed only with V_H MAK33 (Fab-SLC/Fab-LC and V_H MAK33 \pm SLC components) using a streptavidin ELISA format with biotinylated creatine kinase immobilized on the plate (Methods, ELISA), and we no longer have the antigen reagent available to extend the measurements at this time. We have revised the text to state explicitly that the creatine kinase binding data pertain to V_H MAK33 and that testing with V_H 1HEZ remains to be determined in future work.

Reviewer #2 (Remarks to the Author):

In this manuscript, König et al. examined the assembly of the pre-B cell receptor in vitro using purified proteins and in HEK293 cells by transient transfection. The pre-B cell receptor, whose crystal structure was reported in 2007, comprises a surrogate light chain (SLC) covalently paired with an antibody heavy chain. The SLC consists of two non-covalently associated proteins, VpreB and $\lambda 5$. The authors show that VpreB, which is unfolded in isolation, adopts its native structure upon interaction with $\lambda 5$. This process (β -strand complementation) requires a β -strand from I5 to provide the missing ninth β -strand of VpreB. In addition, folding of CH1 requires interaction with $\lambda 5$. The biochemical analysis of the assembly process is technically well done and the results largely justify the authors' conclusions. However, it is not quite accurate to claim that “little is known about the assembly process of VpreB and $\lambda 5$ ” (p. 4, lines 4-5). In fact, quite a lot was already known, as evident from the references cited. For example, β -strand complementation by $\lambda 5$ was described in refs. 29 and 30. Nevertheless, the present work provides the most complete picture to date of pre-B cell receptor assembly.

We thank the reviewer for the positive comments on our study. We changed the sentence about what is known on SLC association.

Points to address:

p. 5, 3rd paragraph. Individual SLC proteins for biophysical analysis were produced by in vitro

refolding from inclusion bodies. CD and NMR analyses indicate that isolated VpreB is unfolded. However, finding appropriate conditions for in vitro folding of proteins is often not straightforward. That is to say, failure to refold a protein in vitro does not necessarily mean that the protein is unfolded when produced naturally within a cell in an environment that includes chaperones, etc. The authors should include this caveat. Having said that, it is certainly striking that addition of $\lambda 5\beta$ peptide to isolated VpreB induces folding of VpreB, as indicated by NMR.

As suggested by the reviewer, we have revised the paragraph on page 5 to clarify the properties of VpreB and to state more clearly its dependence on $\lambda 5$ or the $\lambda 5\beta$ peptide for folding (p.5):

“The far-UV CD spectrum of $\lambda 5$ (Fig. 1d) revealed that the isolated protein is characterized by β -strands and unfolded segments consistent with its structure in the SLC complex²¹ and the presence of an unstructured extension. In contrast to the structure of VpreB in the SLC complex, the far-UV CD spectrum of isolated VpreB (Fig. 1d) indicated that the protein is largely unfolded under the conditions tested. In agreement with this notion, NMR spectroscopy revealed that the ¹⁵N-¹H HSQC spectrum of isolated VpreB is characteristic of an unfolded protein (Fig. 1e, red spectrum).”

And in the discussion, we added (p. 11):

“Although the in vitro experiments demonstrate that VpreB is unfolded due to the lacking $\beta 6$ strand, its folding state in the cell may be further affected by molecular chaperones until the interaction with $\lambda 5$ occurs.”

Fig. 1c. Figure legend should include Hsc70 and explain that it is a positive control. In medium, there are two bands for VpreB in the $\lambda 5$ + VpreB lane. What is the lower MW band, which does not appear (or just barely) in the VpreB alone lane? Is it another glycoform of VpreB?

We thank the reviewer for this observation. We have revised the legend of Fig. 1c to include Hsc70 and to clarify its role as a control for lysed cells. Regarding the two VpreB bands observed in the $\lambda 5$ + VpreB lane of the medium fraction, the lower molecular weight band most likely corresponds to a non-glycosylated form of VpreB, as also shown in Supplementary Fig. 1a in deglycosylation experiments. The revised legend now reads (p. 25):

“Expression and secretion of V5-tagged $\lambda 5$, FLAG-tagged VpreB, and both proteins together in HEK293T cells. Hsc70 was included as a positive control for lysed cells. In the $\lambda 5$ + VpreB lane of the medium fraction, two bands are observed for VpreB. The lower molecular weight band likely represents a non-glycosylated VpreB (see Supplementary Fig. 1a)”

Fig. 2d. Figure legend is incomplete. How exactly were the SPR measurements done? What was immobilized? What was the analyte? Are the concentrations for the $\lambda 5\beta$ peptide? The description of SPR in Materials and Methods does not answer these specific questions.

We thank the reviewer for pointing this out. We have added a description of the experiment in the Methods section.

Revised Method (p. 20 - 21):

For the SPR experiments monitoring binding between $\lambda 5$ and VpreB variants, VpreB was immobilized on a CM5 sensor chip, and purified $\lambda 5$ was injected as analyte at concentrations ranging from 0.002 - 4 μ M. For the SPR experiments between CH1 and SLC variants, CH1 was immobilized on a CM5 sensor chip, and purified SLC variants were injected as analytes at concentrations ranging from 0.002 to 50 μ M, depending on the variant used. For the SPR experiments monitoring binding between the VH domain and SLC variants, VH 1HEZ or VH MAK33 was immobilized on a CM5 sensor chip via amine coupling, and purified VpreB, $\lambda 5$, or the SLC were injected as analytes at concentrations ranging from 0.003 – 10 μ M, depending on the variant used.

Binding curves were corrected by subtraction of buffer and reference flow cell signals (binding to control flow cells and running buffer) and fitted to a 1:1 binding model. Sensorgrams were analyzed using the Biacore X100 Evaluation Software (Cytiva Sweden AB, Uppsala, Sweden) to obtain the equilibrium dissociation constant (KD), association rate constant (ka), and dissociation rate constant (kd).

p. 6, 3rd paragraph. In AUC, isolated VpreB appears to sediment as a homodimer. How is this possible if isolated VpreB is unfolded? In looking at the sedimentation profile of isolated VpreB in Fig. 2f, the profile is very broad compared to those of the other samples, which we know are folded. Very likely the broad VpreB profile reflects non-specific aggregation rather than discrete dimerization. The authors should revise their text accordingly.

This point has also been addressed in our response to Reviewer 1 (see response to Reviewer 1, comment 5). In short: indeed, this is puzzling, however, the assembly is compatible with disorder(<https://www.nature.com/articles/nature25762>). We have revised the manuscript to address this more clearly (p. 6):

“For isolated VpreB, the sedimentation coefficient was also ~ 3.2 S; however, the broad profile compared to folded controls suggests that this signal arises from a heterogeneous mixture of oligomeric species in the range of dimers to tetramers. This interpretation is consistent with previous reports describing non-native dimeric or higher-order VpreB assemblies formed under certain in vitro conditions³⁰⁻³².”

Fig. 3g. Figure legend should explain labeling of samples. Presumably, $\lambda 5DU$ means $\lambda 5$ with URL deleted and VpreBDU means VpreB with UR deleted. Also, for comparison of intensities, the immunoblots should include $\lambda 5$ and VpreB with URs intact.

We thank the reviewer for this suggestion. We have clarified the meaning of the abbreviations in the legend and added a reference to the wild-type controls, which are shown in Supplementary Fig. 3a (p. 26):

“Expression and secretion of murine Fd-MAK33 with ΔUR variants of $\lambda 5$ and VpreB in HEK293T cells. $\lambda 5DU$ denotes $\lambda 5$ with its UR deleted, and VpreBDU denotes VpreB with its UR deleted. For comparison, the expression and secretion of murine Fd-MAK33 alone and in combination with wild-type $\lambda 5$, VpreB, and the complete SLC are shown in Supplementary Fig. 3a.”

Fig. 4b. Figure legend does not adequately describe immunoprecipitations in the two panels.

We agree and have expanded the legend to describe the input and immunoprecipitation panels in more detail (Fig. 4b, p.26):

“Immunoprecipitation of murine Fd-MAK33 with $\lambda 5$, VpreB and the SLC in HEK293T lysates. Left: Input samples showing expression of Fd-MAK33-HA, $\lambda 5$ -V5, and VpreB-FLAG, detected by the indicated antibodies. Hsc70 served as a loading and secretion control. Right: Immunoprecipitation with an anti-HA antibody to isolate Fd-MAK33-HA, followed by immunoblotting for HA, V5, and FLAG to assess co-immunoprecipitation of $\lambda 5$ and VpreB.”

Fig. 5c-e. Same comments about SPR as above for Fig. 2d.

Response:

We thank the reviewer for pointing this out. This issue has been addressed in our response to the related comment above. In short, the method description for SPR was extended.

p. 9, 3rd paragraph. Results of ELISA assays should be confirmed by SPR using immobilized creatine kinase. The authors have all necessary reagents to make these measurements.

We thank the reviewer for this valuable suggestion. We attempted to perform SPR measurements using immobilized creatine kinase to confirm the ELISA results. However, despite multiple optimization attempts, including varying immobilization strategies, buffer conditions, and protein concentrations, we were unable to obtain stable and reproducible binding signals. We suspect that this may be due to a loss of creatine kinase activity upon immobilization. Therefore, we were not able to include SPR data on CK binding in the present manuscript.

Minor:

On p. 2, 1st line, (REF) should give reference number.

In Fig. 3 legend, panels b and e should be panels b and d.

Thanks for detecting this! Both minor points have been addressed. The reference number has been added and the panel labelling has been changed

POINT TO POINT REPLY TO THE REVIEWER COMMENTS

Reviewer #1 (Remarks to the Author):

The authors have now familiarised them with and cite the existing literature concerning the pre-BCR structure and function.

They also have now addressed most of my comments to improve their MS

We thank reviewer for their positive and constructive assessment of the revised manuscript. We are grateful for the acknowledgement that the relevant literature concerning pre-BCR structure and function has now been appropriately incorporated and that the previous comments have been addressed.

Reviewer #2 (Remarks to the Author):

The authors have responded satisfactorily to the previous critiques.

We thank reviewer for their positive evaluation and for confirming that the responses to the previous critiques were satisfactory.